# Shared inflammatory glial cell signature after stab wound injury, revealed by spatial, temporal, and cell-type-specific profiling of the murine cerebral cortex

Christina Koupourtidou [1,2,14], Veronika Schwarz[1,2,14], Hananeh Aliee[3], Simon Frerich [2,4], Judith Fischer-Sternjak[5,6], Riccardo Bocchi [5,6,7], Tatiana Simon-Ebert[5,6], Xianshu Bai [8,9], Swetlana Sirko [5,6], Frank Kirchhoff [8,9,10], Martin Dichgans [4,11,12], Magdalena Götz [5,6,11], Fabian J. Theis [3,13] & Jovica Ninkovic [1,6,11] ✉

Traumatic brain injury leads to a highly orchestrated immune- and glial cell response partially responsible for long-lasting disability and the development of secondary neurodegenerative diseases. A holistic understanding of the mechanisms controlling the responses of specific cell types and their crosstalk is required to develop an efficient strategy for better regeneration. Here, we combine spatial and single-cell transcriptomics to chart the transcriptomic signature of the injured male murine cerebral cortex, and identify specific states of different glial cells contributing to this signature. Interestingly, distinct glial cells share a large fraction of injury-regulated genes, including inflammatory programs downstream of the innate immune-associated pathways Cxcr3 and Tlr1/2. Systemic manipulation of these pathways decreases the reactivity state of glial cells associated with poor regeneration. The functional relevance of the discovered shared signature of glial cells highlights the importance of our resource enabling comprehensive analysis of early events after brain injury.

Traumatic brain injury (TBI) defined as acute brain insult due to an external force, such as the direct impact of a penetrating object or acceleration/deceleration force-induced concussions, affects people of all ages and is among the major causes of death and disability[1,2]. TBI-induced primary damage leads to neuronal and glial cell death, axonal damage, edema, and disruption of the blood-brain barrier (BBB)[3,4]. The initial insult is followed by progressive secondary damage, which further induces neuronal circuit dysfunction, neuroinflammation,

[1]Chair of Cell Biology and Anatomy, Biomedical Center (BMC), Faculty of Medicine, LMU Munich, Planegg-Martinsried, Germany. [2]Graduate School of Systemic Neurosciences, LMU Munich, Munich, Germany. [3]Institute of Computational Biology, Helmholtz Zentrum München-German Research Center for Environmental Health, Neuherberg, Germany. [4]Institute for Stroke and Dementia Research, LMU University Hospital, LMU Munich, Munich, Germany. [5]Chair of Physiological Genomics, Biomedical Center (BMC), Faculty of Medicine, LMU Munich, Planegg-Martinsried, Germany. [6]Institute of Stem Cell Research, Helmholtz Zentrum München-German Research Center for Environmental Health, Neuherberg, Germany. [7]Department of Basic Neurosciences, University of Geneva, Geneva, Switzerland. [8]Molecular Physiology, Center for Integrative Physiology and Molecular Medicine, University of Saarland, Homburg, Germany. [9]Center for Gender-specific Biology and Medicine (CGBM), University of Saarland, Homburg, Germany. [10]Experimental Research Center for Normal and Pathological Aging, University of Medicine and Pharmacy of Craiova, 200349 Craiova, Romania. [11]Munich Cluster for Systems Neurology SYNERGY, LMU Munich, Munich, Germany. [12]German Centre for Neurodegenerative Diseases, Munich, Germany. [13]Department of Mathematics, Technical University of Munich, Munich, Germany. [14]These authors contributed equally: Christina Koupourtidou, Veronika Schwarz. ✉e-mail: jovica.ninkovic@helmholtz-munich.de

oxidative stress, and protein aggregation. These cellular changes have been associated with prolonged symptom persistence and elevated vulnerability to additional pathologies, including neurodegenerative disorders[4,5]. TBI-induced pathophysiology evolves through a highly orchestrated response of all resident glial cell types (microglia, oligodendrocytes, and their precursor cells as well as astrocytes) with peripherally derived infiltrating immune cell populations[3].

After central nervous system (CNS) insult, brain-resident microglia are rapidly activated and change their morphology to a hypertrophic, ameboid morphology[6]. Reactive microglia proliferate, polarize, extend their processes, and migrate to the injury site[3,7]. Similarly, oligodendrocyte progenitor cells (OPCs), known as NG2 glia, display rapid cellular changes in response to damage, including hypertrophy, proliferation, polarization, and migration toward the injury site[8–11]. Astrocytes also react to injury with changes in their morphology, gene expression, and function in a process referred to as "reactive astrogliosis"[8,12,13]. Reactive astrocytes are characterized by upregulation of intermediate filaments, such as glial fibrillary acidic protein (GFAP), nestin, and vimentin[13–15]. Very recently, it was shown that acutely injured myelinating oligodendrocytes started expressing astroglial gene programs and differentiated via a transient bipotential AO cell phenotype with astroglial and oligodendroglial properties into bona fide astrocytes[16].

In response to stab wound injury (SWI), astrocytes, in contrast to microglia and OPCs, do not migrate to injury sites, and only a small proportion of astrocytes near blood vessels (juxtavascular astrocytes) proliferate[17]. These initial responses facilitate the formation of a glial border between intact and damaged tissue[12,17,18], which is necessary not only to restrict the damage[12,18–20], but also to promote axonal regeneration and circuit restoration[12,20–22]. However, adequate border establishment requires well-orchestrated glial cell reactions in relative distance to the injury site. For example, the distance of astrocytes and OPCs from the injury site has been demonstrated to shape their reactive state[11,17,23]. Furthermore, cross-communication among cell types in several pathological conditions[24–26], including TBI[27,28], has been reported to determine cell reactivity states. For example, in neuroinflammatory conditions, reactive microglia induce astrocyte neurotoxicity[29]. Moreover, proliferating astrocytes regulate monocyte invasion[27], whereas BBB dysfunction alters astrocyte homeostasis and contributes to epileptic episodes[30,31]. However, the scope of most studies has been restricted to single cellular populations or the interaction of two cell types at most, a detailed investigation of cellular crosstalk after TBI remains lacking. To obtain a holistic understanding of the cellular responses after brain injury, simultaneous examination of multiple cell types in the injury milieu is critical.

Here, we transcriptomically profile SWI-induced cell reactivity at spatial and single-cell resolution to identify interconnected pathways regulating glial border formation in an unbiased manner. Our data provide insights into the spatial, temporal, and single-cell responses of multiple cell types, and reveal common injury-induced, shared glial signature involving the Toll-like receptor 1/2 (Tlr1/2) and Chemokine receptor 3 (Cxcr3) signaling pathways.

## Results

### Stab wound injury elicits a localized transcriptomic profile in the murine cerebral cortex

TBI induces coordinated cellular reactions leading to glial border formation and isolation of the injury site from adjacent healthy tissue[12]. Importantly, the TBI-induced cellular response is dependent on the distance to the injury site[9,17]. For unbiased identification of regulatory pathways leading to specific spatially defined reactions of glial cells associated with glial border formation, we used spatial transcriptomic (stRNA-seq, 10x Visium). SWI was induced at the border between the motor and somatosensory cortex in both hemispheres, harming only the gray matter[32]. Because our main focus was on examining the

injury-induced changes in the cerebral cortex, we collected brain sections containing the following areas: cortex (CTX), white matter (WM), and hippocampal formation (HPF), as identified on the basis of the Allen brain atlas (Fig. 1a, b). This allowed us to position two brain sections containing the dorsal telencephalic regions on a single capture area (Supplementary Fig. 1a, b). This approach provided the advantage to investigate the expression of a multitude of genes from all cell types at the injury site and to examine their dynamics as a function of distance from the injury site. The primary impact initiates a cascade of processes, which involve the reactions of glial cells and infiltrating or resident immune cells[23,33]. To capture the response of infiltrating immune cells, which peaks at 3 days post-injury (dpi)[27], and glial cells, which peaks at 2–5 dpi[11,13,27], we performed stRNA-seq at 3 dpi. Injury-induced alterations were determined by comparison of stab-wounded brain sections to corresponding intact sections (Fig. 1c, Supplementary Fig. 1a, b).

Notably, we were able to identify clusters corresponding to specific anatomical structures, e.g., cluster II expressing genes characteristic of cortical layer 2/3 neurons; cluster VIII expressing genes representing layer 4 neurons; and cluster I and cluster IV expressing genes identifying layer 5 and layer 6 neurons, respectively[34] (Supplementary Fig. 1c, d, Supplementary Tables 1, 2). Importantly, the global cortical layer patterning was not affected by the injury, because we also observed similar gene expression patterns identifying the same neuronal layers in the injured brain sections (Supplementary Fig. 1d) in line with previous studies[35]. However, beyond clusters characterizing individual anatomical structures, we identified an injury-induced cluster, cluster VI, localized around the injury core (Fig. 1b, c, Supplementary Fig. 1b). Interestingly, cluster VI was distributed throughout cortical layers 1–5 and was absent in the intact brain sections (Fig. 1c, Supplementary Fig. 1a, b). Cluster VI was characterized by a specific transcriptomic signature with enrichment in genes associated with reactive astrocytes[12,36–38] (*Gfap, Lcn2, Serpina3n, Vim, Lgals1, Fabp7,* and *Tspo*) and microglia[39] (*Aif1, Csf1r, Cd68,* and *Tspo*), which have been associated with CNS damage[12] (Fig. 1d, Supplementary Fig. 1c, Supplementary Data 1).

To obtain insight into the regulated processes within cluster VI, we performed Gene Ontology (GO) enrichment analysis of significantly upregulated genes (pval < 0.05, $\log_2$ fold change >1) in this cluster compared with all other clusters. The over-represented biological processes (BP) were associated with immune response and angiogenesis, whereas the molecular function (MF) and cellular components (CC) indicated changes in genes associated with the extracellular matrix (Fig. 1e, Supplementary Data 1). Notably, the above-mentioned processes have been reported to drive glial reaction in response to brain injury and to facilitate glial border formation[20]. Furthermore, processes associated with phagocytosis (phagocytotic vesicles) were enriched in cluster VI (Fig. 1e, Supplementary Data 1), in line with the previously described importance of phagocytotic processes in the context of brain injuries[40]. To confirm the unique injury-induced expression profile and hence the presence of cluster VI, we validated the expression of the cluster VI genes *Serpina3n, Lcn2,* and *Cd68* at the RNA and protein levels. Indeed, the selected candidates were specifically expressed around the injury core, as predicted by our stRNA-seq analysis (Fig. 1f, g), and these expression patterns were also observed at the protein level (Fig. 1h). Although these selected genes were enriched in cluster VI, they displayed unique expression patterns within the injury-induced cluster. Specifically, *Serpina3n* was expressed more broadly than *Lcn2*, whereas *Cd68* expression was predominantly confined to the immediate injury core (Fig. 1f, g).

To comprehensively determine different expression profiles between the injury core and the perilesional area, we conducted spatial gradient analysis using the SPATA2 analysis pipeline[41]. This allowed us to visualize individual genes and gene set expression patterns as a function of the distance from the injury core (Fig. 2a). For this purpose, we segregated the perilesional area around the injury core into 13

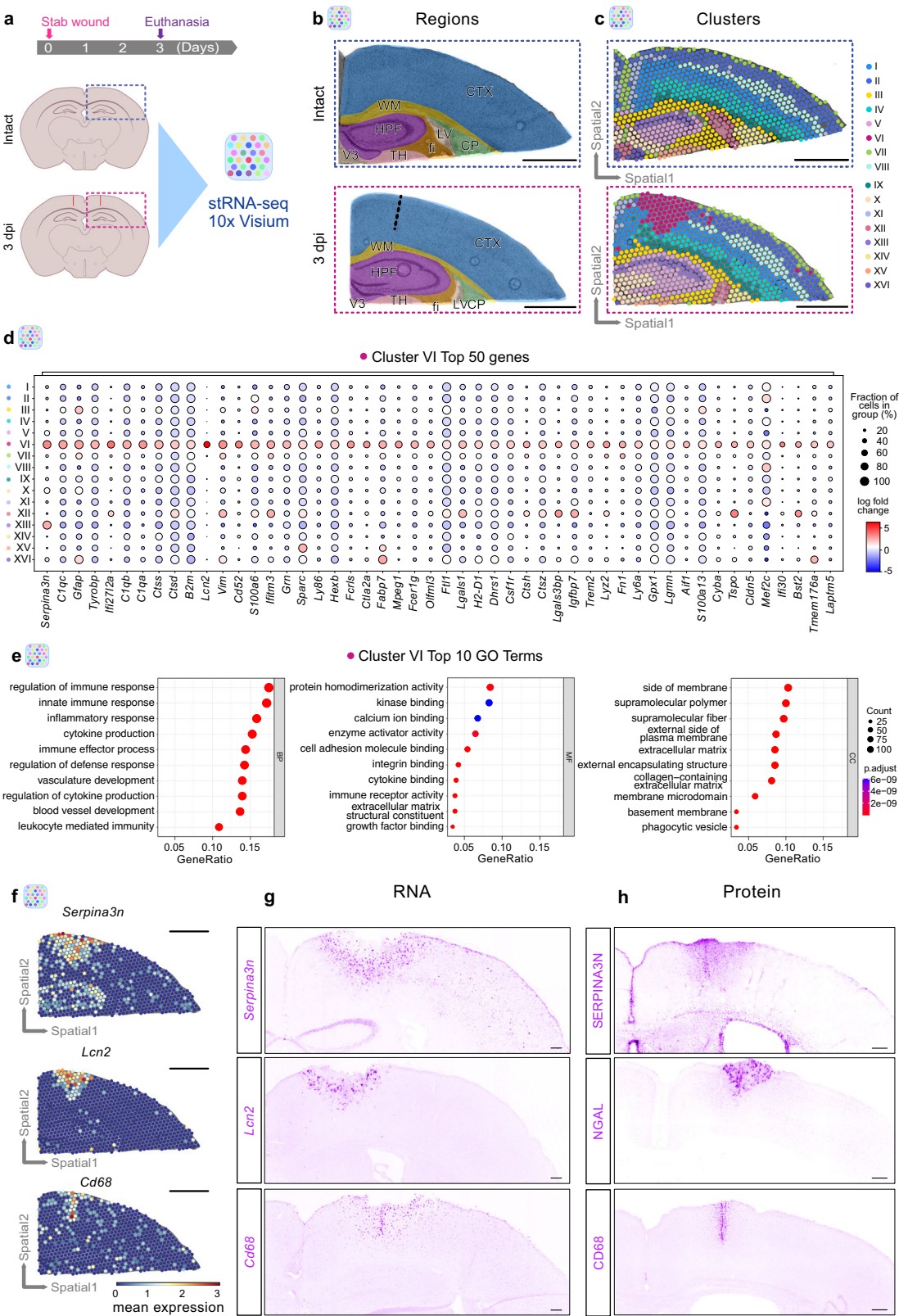

concentric circles (Fig. 2a). In addition, we excluded all subcortical clusters from the analysis and correlated the gene expression profiles along the spatial gradient to a variety of pre-defined models (further details in Methods). To reveal the differences between the injury core and perilesional areas, we focused on the genes with "descending" (enriched at the injury core) (Fig. 2b) and "ascending" (depleted at injury core) expression profiles (Supplementary Fig. 1e). As expected,

all top descending genes were highly enriched at the injury core. However, in the perilesional area (as defined by the border of cluster VI; ~0.5 mm distance from the injury core) some of these genes displayed unique descending rates (Fig. 2b). We observed heterogeneous expression patterning, ranging from injury core-confined expression (e.g., *Alox5ap* and *Rplp0*) to wide-ranging expression (e.g., *Fth1* and *Gfap*) reaching far from the cluster VI border (Fig. 2b). Of note, gene

**Fig. 1 | Spatially resolved transcriptomic changes induced by stab wound injury. a** Experimental scheme to conduct spatial transcriptomics in intact and stab wound-injured mouse cerebral cortices (3 dpi). Brains were manually resected, and selected areas highlighted in blue or red dashed boxes were positioned on 10x Visium capture areas. **b** Brain sections of both conditions contain cortical, hippocampal, and white matter regions. The black dashed line indicates the injury core. **c** Clustering of gene expression data on spatial coordinates based on highly variable genes and subsequent dimensionality reduction. **d** Dot plot illustrating the expression of the 50 most enriched genes in the injury-induced cluster VI. **e** Dot plots depicting GO terms over-represented in cluster VI significantly enriched genes (pval < 0.05, log$_2$ fold change >1, method t-test overestimated_variance).

Significance of GO terms was determined by performing GO enrichment analysis on gene sets. Spatial transcriptomic analysis (**a**–**e**) is based on $n = 1$ animal per condition over 1 experiment, with two sections being captured. **f** Gene expression of cluster VI-enriched genes *Serpina3n*, *Lcn2*, and *Cd68* in spatial context. **g, h** Images depicting expression of *Serpina3n*, *Lcn2*, and *Cd68* at the RNA (**g**) and protein level (**h**) in stab wound-injured cerebral cortices at 3 dpi ($n = 3$ animals). All images are full z-projections of confocal z-stacks. Scale bars: **b, c, f:** 1 mm, **g, h:** 150 μm. stRNA-seq spatial transcriptomics, CTX cerebral cortex, WM white matter, HPF hippocampal formation, LV lateral ventricle, CP caudoputamen, V3 third ventricle, TH thalamus, fi fimbria, dpi days post-injury, BP biological processes, MF molecular functions, CC cellular components, GO gene ontology.

sets associated with the immune response and inflammation were particularly enriched at the injury core (Fig. 2c, Supplementary Data 2), whereas gene sets associated with neuronal and synaptic activity were enriched only in the perilesional areas (Supplementary Fig. 1f, Supplementary Data 2). Similarly, to the descending genes, the ascending genes exhibited relatively divergent expression profiles in the perilesional areas (Supplementary Fig. 1e). However, approximately 50% of all top 25 ascending genes were associated with mitochondrial functions, in contrast to the descending genes. These mitochondrial genes exhibited almost identical expression profiles in the perilesional area (Supplementary Fig. 1e), thus supporting prior findings that brain insult disrupts normally well-regulated mitochondrial function in a coordinated manner[42–44].

In summary, with our spatial gene expression analysis, we identified well-defined anatomical structures as well as an injury-specific cluster characterized by angiogenesis and immune system-associated processes, including phagocytosis. Furthermore, by using spatial gradient analysis, we highlighted injury-induced heterogeneous gene expression profiles in the perilesional area.

## Multiple cellular states contribute to injury-induced local transcriptome profiles

Although stRNA-seq enables profiling of transcriptomic changes by preserving spatial information, the profile itself is derived from multiple cells, which are captured in each spot (1–10 cell resolution). To assess the cellular composition of the injured area and to identify which cell populations defined the transcriptomic profile of cluster VI, we performed single-cell transcriptomic (scRNA-seq) analysis of stab wound-injured cortices 3 dpi and corresponding areas in the intact cortex, by using a droplet-based approach (10x Chromium) (Fig. 3a). After applying quality control filters, we identified a total of 6322 single cells (Fig. 3b) emerging from both conditions (intact: 2676 cells, 3 dpi: 3646 cells, Fig. 3c, Supplementary Fig 2a), which, on the basis of their gene expression, were distributed among 30 distinct clusters. Through this approach, we identified neuronal and glial clusters, including astrocytes, microglia, and oligodendrocyte lineage cells, in addition to vascular cells, pericytes, and multiple types of immune cells (Fig. 3b, Supplementary Fig. 2a, b, Supplementary Data 3). Additionally, we generated gene expression scores based on established marker genes of well-characterized cell populations in the adult mouse brain (Supplementary Table 3). Indeed, the gene scores exhibited enrichment in the corresponding cellular populations, thus further validating our cluster annotation (Fig. 3d).

Interestingly, by comparing the cell distributions between the intact and injured conditions, we observed that several clusters of immune and glial cells were highly abundant exclusively in the injured brain (Fig. 3c, Supplementary Fig. 2a). The clusters 8_NKT/T cells, 13_Macrophages/Monocytes, 17_DCs, 18_Monocytes, and 22_DCs, for example, appeared primarily after injury and expressed *Ccr2*[6,22] (Supplementary Fig. 2c) in addition to their distinct cell identity markers (Supplementary Fig. 2b, Supplementary Data 3). Microglia clusters that appeared after injury (11_Microglia and 16_Microglia) exhibited high expression of *Aif1* and low expression of the

homeostatic microglia markers *Tmem119* and *P2ry12*[45] (Supplementary Fig. 2c). Similarly, the astrocytic clusters 12_Astrocytes and 23_Astrocytes were present primarily in the injured condition and were characterized by high expression of classical reactive astrocyte markers such as *Gfap* and *Lcn2*[36,38] (Supplementary Fig. 2d). In addition to microglia and astrocytes, the cluster 15_OPCs was present primarily after injury (Supplementary Fig. 2a). Cells from cluster 15_OPCs expressed a combination of genes associated with the cell cycle (G2/M phase, Supplementary Fig 2f, Supplementary Table 4)[46,47] and *Cspg4* (Supplementary Fig. 2e); both hallmarks of Nerve/glial antigen 2 glia (NG2 glia), which rapidly proliferate after brain injury[9].

To elucidate which of these cellular clusters contributed to the injury-specific signature of cluster VI, we mapped the single-cell expression data onto the spatial gene expression dataset (Fig. 3e, f, Supplementary Figs. 3 and 4) by using Tangram[48]. To include the identical anatomical regions regarding the scRNA-seq data acquisition, we restricted the stRNA-seq dataset to the cortical clusters (clusters I, II, IV, VI, VII, VIII, IX, and XI). The probabilistic mapping predicted that several clusters including 11_Microglia, 16_Microglia, 12_Astrocytes, 23_Astrocytes, 13_Macrophages/Monocytes, 18_Monocytes, and 15_OPCs were localized near the injury core (Fig. 3e, Supplementary Fig. 3). In contrast, neuronal clusters 1_Neurons, 2_Neurons, and 24_Neurons, as well as the astrocytic clusters 3_Astrocytes, 5_Astrocytes, 7_Astrocytes, and 9_Astrocytes, displayed decreased representation around the injury site (Fig. 3e, Supplementary Fig. 3). Additionally, we used the H&E images of the stRNA-seq dataset to estimate the number of nuclei within each spot of the capture area, which, in combination with probabilistic mapping, can be used for deconvolution. To some extent, this analysis further associated the above-mentioned clusters with the injury milieu (Fig. 3f, Supplementary Fig. 4a, b). Importantly, glial cells mapped to the injury cluster VI cluster together, as unveiled by the hierarchical clustering (Fig. 3e). This implies a similar response to injury within certain glial cell clusters. It is essential to note, however, that not every cluster of glial cells actively contributes to shaping the injury environment (Supplementary Figs. 3, 4b). Most astrocytic clusters, with the exception of clusters 12_Astrocytes and 23_Astrocytes, did not show enriched mapping at the injury site (Fig. 3e, Supplementary Fig. 3). Similar behavior was detected for the oligodendrocyte clusters 20_MOL and 27_COPs (Supplementary Fig. 3). Notably, our deconvolution analysis supported these observations (Supplementary Fig. 4b). In summary, the combination of stRNA-seq with corresponding scRNA-seq datasets allowed us to identify an injury-specific transcriptional profile exhibiting enrichment of individual glial subpopulations, and subsequent depletion of distinct astrocytic and neuronal clusters.

## Stab wound injury induces common transcriptomic changes in glial cells

Because glial cell reactivity exhibits distinct temporal dynamics in response to injury[11,13,27], we decided to add an additional time point (5 dpi) to our scRNA-seq analysis (Fig. 4a). This experimental design enabled investigation of the transcriptional states of glial cells underlying the observed heterogeneity in glial cell responses.

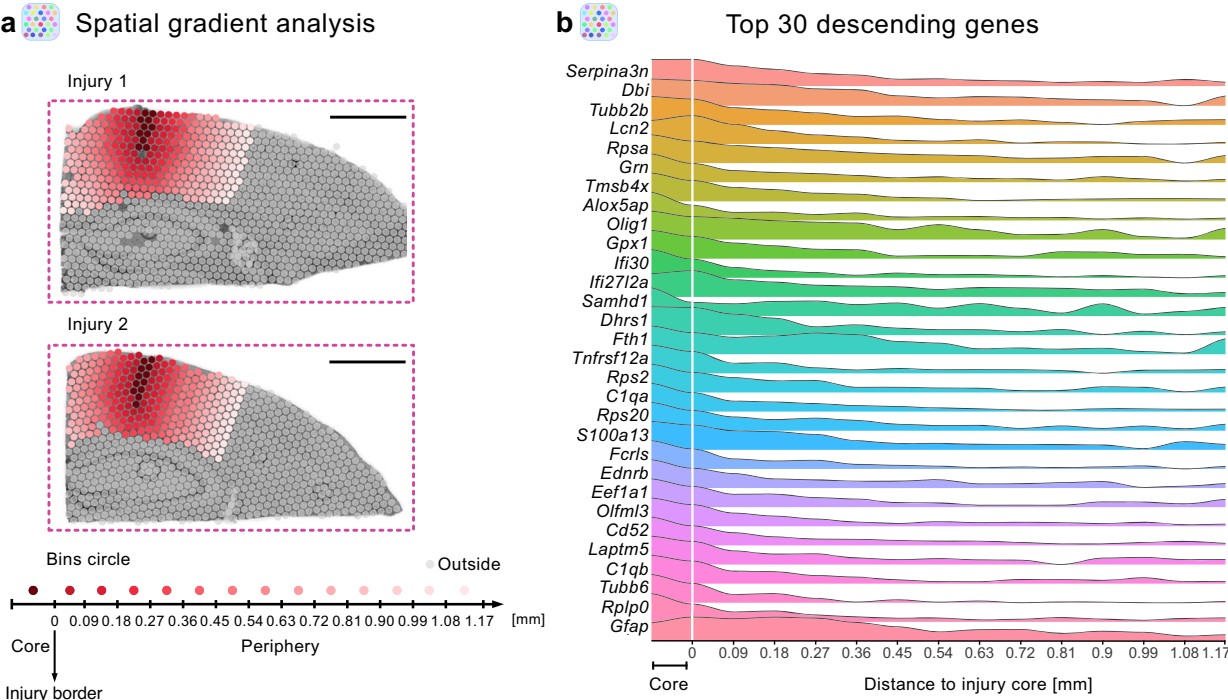

**a** Spatial gradient analysis

**b** Top 30 descending genes

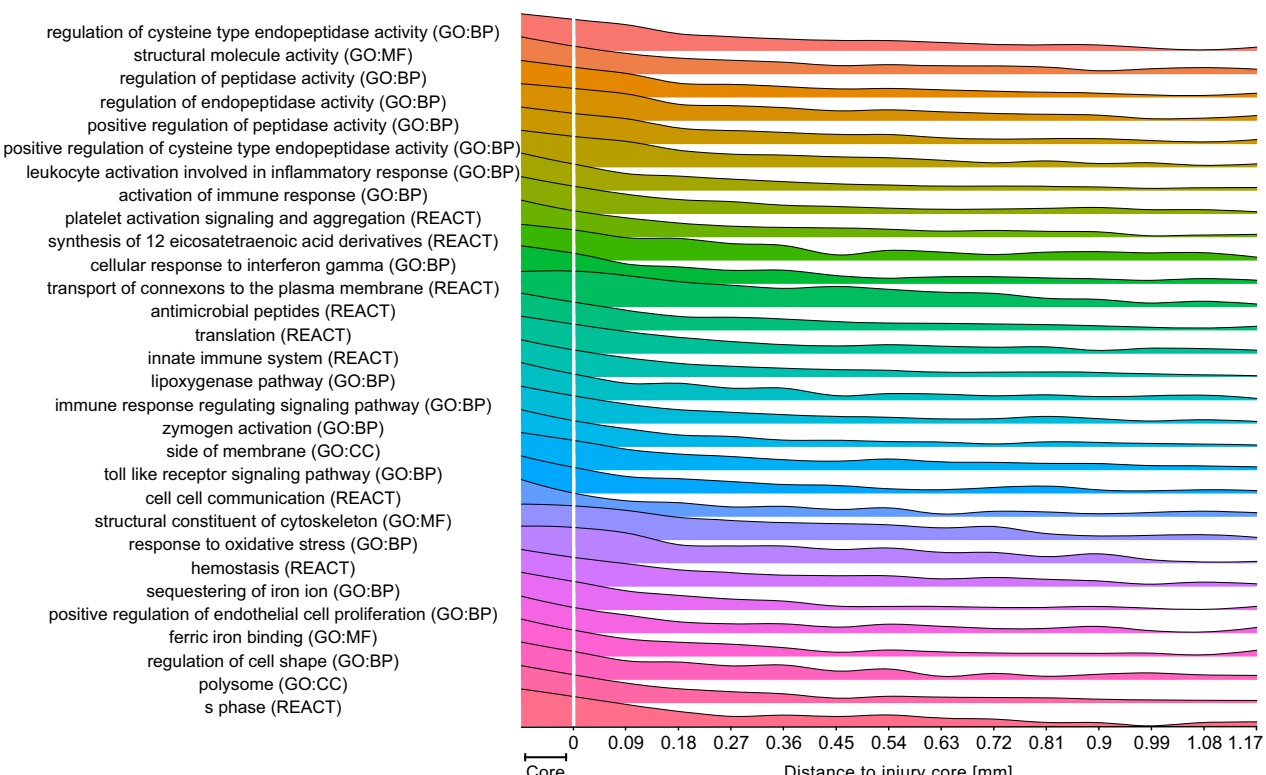

**c** Top 30 descending gene sets

**Fig. 2 | Stab wound injury elicits distinct gene expression patterning along a spatial gradient. a** Paradigm for spatial transcriptomic gradient analysis on stab wound-injured mouse cerebral cortices (3 dpi) using the SPATA2 pipeline. Spatial gradient analysis was conducted only in cortical areas; from the injury core (dark red spots) toward the periphery (light pink) within 13 concentric circle bins. All other areas (gray spots) were neglected. **b** Ridge plot depicting the expression of the 30 most descending genes along the gradient, depicted as mean expression in two injuries (injury 1 and 2). **c** Ridge plot displaying the top 30 most descending gene sets along the gradient, depicted as mean expression in two injuries (injury 1 and 2). stRNAseq data are based on *n* = 1 animal over 1 experiment with two injured hemispheres being captured. Scale bars 1 mm. BP biological processes, MF molecular functions, CC cellular components, GO gene ontology, REACT reactome.

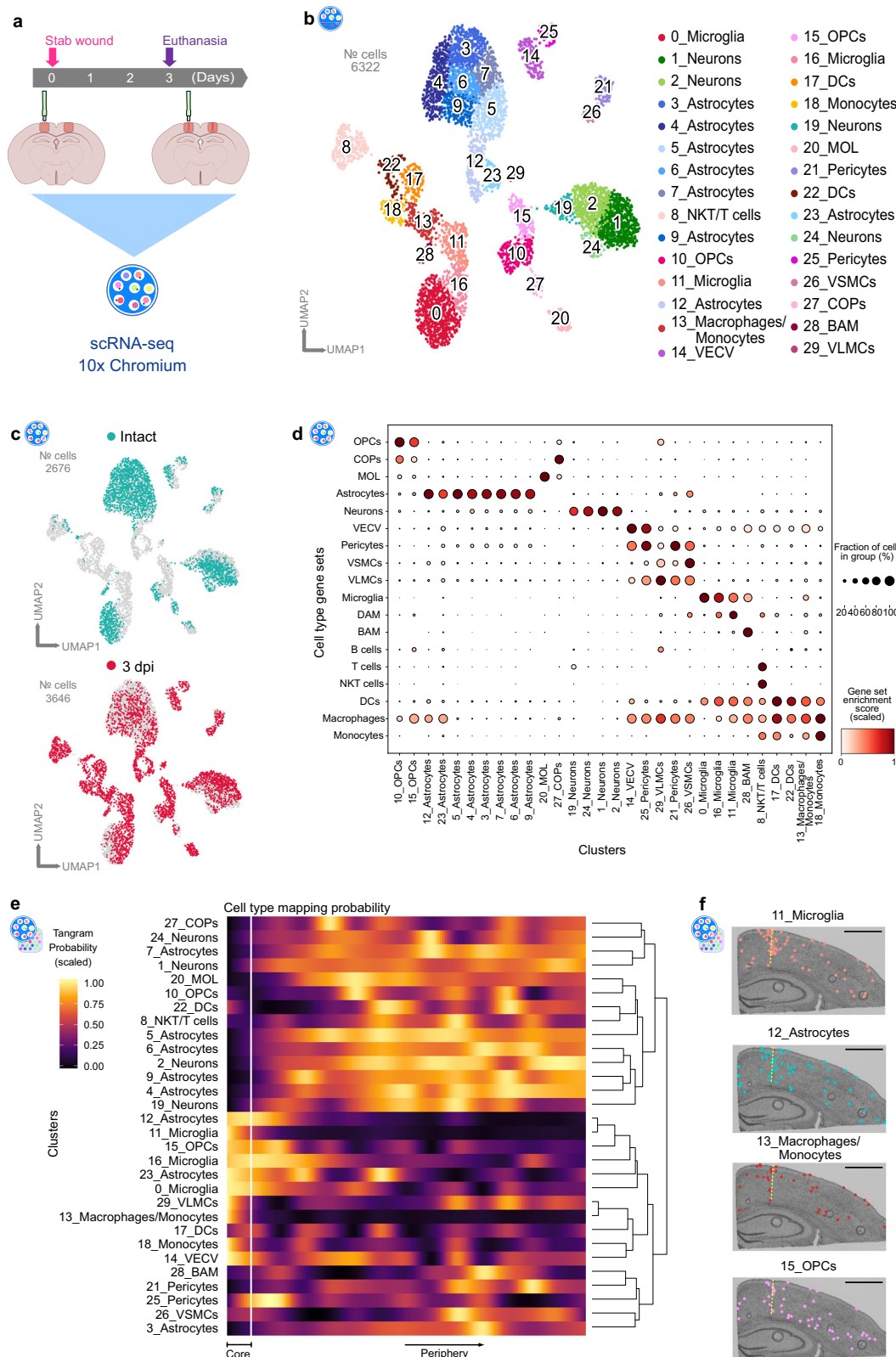

In total, we analyzed 35132 cells (intact: 16964, 3 dpi: 3646, 5 dpi: 14522), which were distributed among 30 clusters (Fig. 4b, Supplementary Fig. 5a–c, Supplementary Data 4). In line with our previous observations (Fig. 3c), we identified several injury-induced clusters, which were formed exclusively by cells originating from the brains of injured animals (Supplementary Fig. 5b). Notably, within this clusters, a fraction of these cells initiates proliferation by entering the

S-phase (Supplementary Fig. 5d), underscoring the intricate cellular responses triggered by injury. However, none of these injury-induced clusters were specific to either the 3 or 5 dpi time point (Supplementary Fig. 5b).

To unravel how each glial population transited from a homeostatic to a reactive state, we focused on individual cell populations. Hence, we further subclustered astrocytes, microglia, and oligodendrocytes

**Fig. 3 | Combination of spatial and single-cell transcriptomics identifies cellular populations contributing to distinct transcriptional responses at the injury site. a** Experimental scheme to conduct single-cell RNA-sequencing of intact and stab wound-injured cerebral cortices (3 dpi) with the 10x Genomics platform. Red masked areas on brain schemes indicate biopsy areas used for the analysis. **b** UMAP plot illustrating 6322 single cells distributed among 30 distinct clusters. Clusters are color-coded and annotated according to their transcriptional identities. **c** UMAP plot depicting the distribution of cells isolated from intact (green) and injured (red) cerebral cortices. **d** Dot plot indicating strong correlation of post hoc cluster annotation with established cell-type-specific gene sets (Supplementary Table 3). **e, f** 3 dpi scRNA-seq cluster localization along the spatial gradient (Fig. 2),

based on probabilistic mapping with Tangram (**e**) and single-cell deconvolution (**f**) in a spatial context. scRNAseq data shown in this figure are derived from *n* = 3 intact animals over 1 experiment (1 scRNAseq libray) and *n* = 3 3dpi animals over 1 experiment (2 scRNAseq libraries). stRNAseq data are based on *n* = 1 animal over 1 experiment with two injured hemispheres being captured. Scale bars in (**f**): 1 mm. scRNA-seq single-cell transcriptomics, UMAP uniform manifold approximation and projection, dpi days post-injury, OPCs oligodendrocyte progenitor cells, COPs committed oligodendrocyte progenitors, MOL mature oligodendrocytes, VECV vascular endothelial cells (venous), VSMCs vascular smooth muscle cells, VLMCs vascular and leptomeningeal cells, DAM disease-associated microglia, BAM border-associated macrophages, NKT cells natural killer T cells, DCs dendritic cells.

(Fig. 4c–e, Supplementary Fig. 6). We identified distinct clusters in each of the investigated populations, which were composed primarily of cells from intact (blue clusters) and injured (orange/red clusters) samples (Fig. 4c–e, Supplementary Fig. 6c, h, m). Additionally, cells originating from the injured samples expressed typical markers of glial reactivity. We identified clusters AG5, AG6, and AG7 as the main populations of reactive astrocytes (Fig. 4c, Supplementary Fig. 6a–e), because these cells expressed high levels of *Gfap*, *Vim*, and *Lcn2*[36,38] (Supplementary Fig. 6e). Microglial clusters MG4 and MG7 displayed high expression of *Aif1* and low expression of the homeostatic markers *Tmem119* and *P2ry12*[45] (Fig. 4d, Supplementary Fig. 6f–j). By subclustering cells belonging to the oligodendrocyte lineage, we were able to identify two populations of OPCs (OPCs1 and OPCs2) (Fig. 4e, Supplementary Fig. 6k–o). Cluster OPCs2 was composed primarily of cells from injured samples (Supplementary Fig. 6m). Of note, we were not able to find a unique marker within the OPCs2 cluster for identifying reactive OPCs (Supplementary Fig. 6o). Importantly, cells from the injury-responding clusters, as identified by our previous deconvolution analysis (11_Microglia, 12_Astrocytes, and 15_OPCs) (Fig. 3f, Supplementary Fig. 4), also mapped predominantly to the glial subclusters evoked by injury (Supplementary Fig. 6d, i, n). Together, these results corroborated the reactive state of these glial subclusters (henceforth referred to as reactive clusters).

By subclustering glial cells, we did not discover any cluster unique to either the 3 or 5 dpi time point. This finding suggests a gradual activation of glial cells in response to injury and therefore contributing to reactive clusters with different abundances (Fig. 4f–h). More specifically, many of the astrocytes at 3 dpi remained present in the homeostatic clusters and were only partially present in the reactive clusters AG6 and AG7, whereas at 5 dpi, most cells were detected in cluster AG5 (Fig. 4f). Microglia, in contrast, displayed a faster transition to reactivity than astrocytes: at 3 dpi, most cells were already localized in the reactive clusters MG4 and MG7. At 5 dpi, however, most of the cells had begun to transition back to the homeostatic state, and a high proportion of cells were present in cluster MG3 (Fig. 4g). Similarly, OPCs reacted rapidly after injury, because at 3 dpi, most cells resided in the reactive cluster OPCs2, whereas only several cells were present in this cluster at 5 dpi (Fig. 4h). These results are in line with prior findings showing that microglia and OPCs rapidly respond to injury, and their reactivity peak ranges from 2 to 3 dpi, whereas astrocyte reactivity peaks at 5 dpi[11,13,27].

The enriched immune system-associated processes around the injury site, as indicated by spatial transcriptomics, and the identification of specific reactive subtypes of glial cells populating the injury environment, prompted the question of whether the inflammatory gene expression might be a unique signature of one specific cell type or a common feature of reactive glia. Therefore, within each glial population (Fig. 4c–e), we extracted the differentially expressed genes (DEGs) of each glial subcluster (pval < 0.05, log$_2$ fold change >1.6 or < −1.6) and compared them among all subclusters (Supplementary Fig. 7a, b, Supplementary Data 5). Interestingly, among all clusters, the highest similarity of upregulated genes was observed between the reactive glial clusters MG4, AG5, and OPCs2, with 141 enriched genes in

common (Supplementary Fig. 7b). This finding suggested that, in response to injury, individual reactive glial clusters might share some cellular programs. Hence, we performed GO term analysis of the 241 commonly upregulated genes from the comparison of clusters MG4, AG5, and OPCs2, independently of other glial subclusters (Fig. 5a). Most of the commonly regulated processes were associated with cell proliferation (Fig. 5b, Supplementary Data 6), which has been reported to be a shared hallmark of glial cell reactivity[13,36]. Moreover, we identified processes associated with innate immunity (Fig. 5b, Supplementary Data 6) and numerous genes associated with the type I interferon signaling pathway (*Ifitm3*, *Ifit3*, *Bst2*, *Isg15*, *Ifit3b*, *Irf7*, *Ifit1*, *Ifi27l2a*, *Oasl2*, and *Oas1a*) (Fig. 5c) as well as *Cxcl10*, a ligand activating the Cxcr3 pathway[49]. Indeed, by using RNAscope, we confirmed that *Cxcl10*, *Oasl2*, and *Ifi27l2a* were expressed by a subset of microglia, astrocytes, and OPCs (Supplementary Fig. 8a–t). Notably, the expression of these innate immunity-associated genes was clearly restricted to distinct glial subpopulations, since not all glial cells expressed these markers (Fig. 5d, Supplementary Fig. 8b–t). Additionally, the upregulation of the innate immunity-associated genes (Fig. 5c) is specifically confined to the proximal injury site (Supplementary Fig. 10f) and is a unique feature of glial cells because neurons never expressed these genes, whereas in vascular cells they were present at low levels even in the intact brain (Fig. 5d, e).

We further sought to explore whether the shared inflammatory signature could be a consistent feature of reactive glia in various brain pathologies. Thus, we utilize publicly available scRNA datasets from mild FPI[50] (fluid percussion injury) (Fig. 5f–k) and MCAO[51] (middle cerebral artery occlusion) mouse models (Supplementary Fig. 9a–e), alongside a database documenting the response of astrocytes to systemic lipopolysaccharide (LPS) injection[38] (Supplementary Fig. 9f, g). In the FPI model, no expression of the shared inflammatory signature was detected in glial cells (Fig. 5f, h). This could be attributed to differences in time points and/or the absence of the reactive population of microglia (*Aif1* high/*P2ry12* low) (Fig. 5i) and astrocytes (*Gfap* and *Lcn2* positive) (Fig. 5j) following FPI. In addition, the detected glia populations in the FPI dataset displayed greater similarity in gene expression with the homeostatic populations in our analysis than with the reactive ones (7_Astrocytes, 11_Microglia, and 12_Astrocytes) (Fig. 5k). Conversely, in the stroke model, we could identify the signature in astrocytes and microglia. Of note, OPCs were not retrieved in the MCAO model thus we could not check their reaction to pathology (Supplementary Fig. 9a–e). It is noteworthy that all datasets showed expression of the shared inflammatory signature in endothelial/vascular cells, corroborating the expression already detected in the intact samples of our dataset (Fig. 5h, Supplementary Fig. 9c). Interestingly, a high proportion of the shared inflammatory marker genes (e.g., *Oasl2* and *Ifit1*) were expressed exclusively in the astrocytic cluster 8 (Supplementary Fig. 9f) emerging after LPS treatment[38]. Astrocytes of cluster 8 were classified in the above-mentioned study as reactive astrocytes in a sub-state capable of rapidly responding to inflammation. Notably, not all inflammatory genes were detected in LPS-induced reactive astrocytes, thus indicating that only a portion of the signature was retained (Supplementary Fig. 9g). This finding strongly

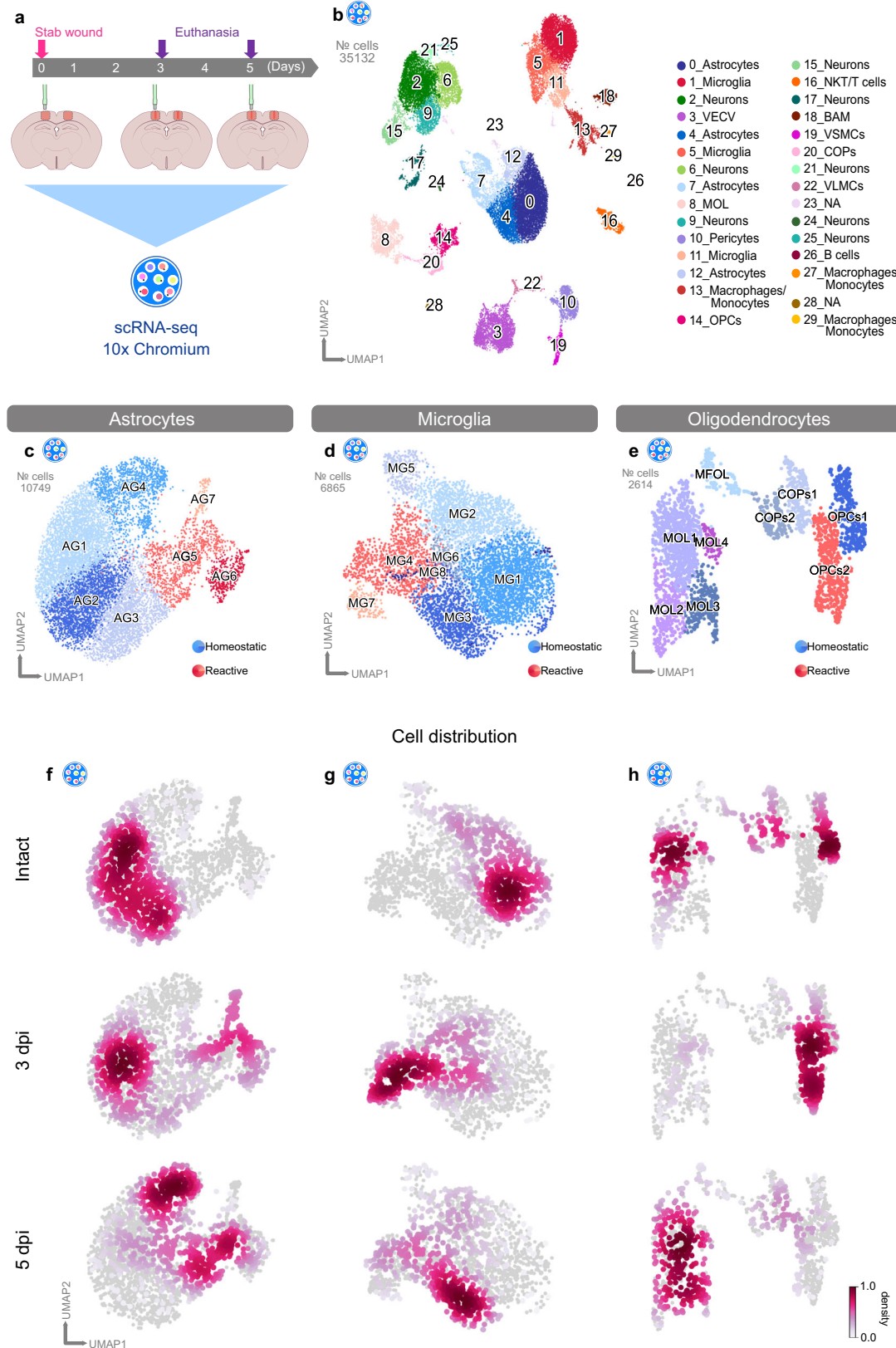

emphasizes the need for a holistic cellular view of brain pathologies to identify therapeutical targetable pathways. Our findings revealed a common inflammatory signature present in a subset of reactive glial cells in response to SWI. Moreover, this shared inflammatory signature is partially preserved in astrocytes and microglia in different brain pathologies.

## Regulation of injury-induced innate immune responses via the Cxcr3 and Tlr1/2 pathways

On the basis of the shared regulation of the innate immunity pathways after brain injury, including the expression of components of the CXC chemokine receptor 3 (Cxcr3) and Toll-like receptor 2 (Tlr2) pathways (Supplementary Fig. 10a) alongside the DAMP signature as a potential

**Fig. 4 | Stab wound injury induces defined transcriptional changes in glial subpopulations. a** Experimental scheme for single-cell RNA-sequencing of intact and stab wound-injured cerebral cortices (3 and 5 dpi) with the 10x Chromium platform. Red masked areas on brain schemes indicate biopsy areas used for the analysis. **b** UMAP embedding of integrated and batch-corrected single-cell transcriptomes of 35132 cells distributed among 30 distinct clusters. Clusters were color-coded and annotated on the basis of their transcriptional identities. **c**–**e** UMAPs depicting subclustering of astrocytes (7 clusters) (**c**), microglia (8 clusters) (**d**), and oligodendrocytes (9 clusters) (**e**). Cells were further assigned to homeostatic (blue) or reactive (red) clusters according to cell origin (Supplementary Fig. 6) and distinct marker expression. **f**–**h** UMAPs illustrating cell distributions at all time points (intact, 3 dpi, and 5 dpi) among subclusters of astrocytes (**f**),

microglia (**g**), and oligodendrocytes (**h**). Data were downsampled to an equal number of cells between time points for each cell type. Data shown in this figure are derived from $n = 12$ intact animals over 3 independent experiments (5 scRNAseq libraries), $n = 3$ 3dpi animals over 1 experiment (2 scRNAseq libraries), and $n = 9$ 5dpi animals over 3 independent experiments (3 scRNAseq libraries). scRNA-seq single-cell transcriptomics, UMAP uniform manifold approximation and projection, dpi days post-injury, OPCs oligodendrocyte progenitor cells, COPs committed oligodendrocyte progenitors, MOL mature oligodendrocytes, VECV vascular endothelial cells (venous), VSMCs vascular smooth muscle cells, VLMCs vascular and leptomeningeal cells, BAM border-associated macrophages, NKT cells natural killer T cells, NA not available.

activator[52] (Supplementary Fig. 10b–e), we investigated the injury-induced transcriptional changes after interference with the Cxcr3 and Tlr1/2 pathways. This investigation stems from our recent findings that Cxcr3 and Tlr1/2 regulate OPC accumulation at injury site in the zebrafish brain[53]. For our study we used the Cxcr3 antagonist NBI 74330[54] and the Tlr1/2 pathway inhibitor CU CPT 22[55] to interfere with the above-mentioned pathways, and performed scRNA-seq analysis at 3 dpi and 5 dpi (henceforth referred to as 3/5 dpi_INH) (Fig. 6a). The specificity of these chemical compounds was validated with a murine knock-out OPC cell line[53]. The data were integrated with our previously acquired datasets (Fig. 4b) from intact (INT) and injured animals (henceforth referred to as 3/5 dpi_CTRL). In total, we analyzed 55405 cells (INT: 16964 cells, 3 dpi_CTRL: 3646 cells, 3 dpi_INH: 4615 cells, 5 dpi_CTRL: 14522 cells, 5 dpi_INH: 15658) distributed among 34 clusters (Fig. 6b, Supplementary Fig. 11a). Notably, with the integration of additional conditions (3 and 5 dpi_INH), and hence a subsequent increase in the total cell number, we did not observe the emergence of new clusters (Supplementary Fig. 11a). Furthermore, even after the integration of the INH datasets, the overall cluster identity was unaffected, as indicated by high similarity scores among the clusters (Supplementary Fig. 11b). Because microglia, astrocytes, and OPCs displayed common innate immune-associated gene expression after SWI (Fig. 5c–e), we sought to investigate the possible influence of Cxcr3 and Tlr1/2 pathway inhibition on microglia, astrocytes, and OPCs by further subclustering the above-mentioned cell types. In each investigated cell population, we again identified distinct clusters containing primarily cells from injured samples (Fig. 6c–e, Supplementary Fig. 11c–e). Of note, these clusters were composed of cells originating from both CTRL and INH samples. These results suggested that the inhibition of Cxcr3 and Tlr1/2 pathways after SWI did not induce new transcriptional states. Instead, the inhibitor treatment resulted in partial downregulation of the shared inflammatory genes (Fig. 5c) in the reactive clusters AG7, MG4, and OPCs2 (Fig. 6f, Supplementary Table 5).

To address transcriptional changes induced by the inhibitor treatment, we performed differential gene expression analysis of each subcluster between CTRL and INH conditions at each time point (pval < 0.05, $\log_2$ fold change >0.7 or $\log_2$ fold change <−0.7). Interestingly, most of the inhibitor-induced changes at 3 and 5 dpi were subcluster specific, because only a few DEGs overlapped (Supplementary Fig. 12a–d). To reveal the biological processes regulated in each glial subcluster (Fig. 6c–e), we used the function compareCluster[56] (clusterProfiler R package) and calculated the enriched functional profiles of each cluster. This function summarized the results into a single object and allowed us to compare the enriched biological processes of all glial subclusters at once. Indeed, by comparing the processes associated with all significantly downregulated genes after treatment at 3 dpi, we identified many programs associated with the innate immune response, which were shared among several glial populations, including reactive astrocytes (clusters AG5, AG6, AG7, and AG8), microglia (clusters MG4 and MG6), and OPCs (cluster OPCs2) (Fig. 6g, Supplementary Data 7). Interestingly, although still downregulated at 5 dpi, these immune response-associated processes

were no longer shared among different glial populations (Fig. 6h, Supplementary Data 7). In contrast, biological processes induced by the inhibitor treatment were cluster-specific, independently of the analysis time point (Supplementary Fig. 12e, f). Together, our scRNA-seq analysis findings indicated that the Cxcr3 and Tlr1/2 signaling pathways regulate similar processes in initial activation (3 dpi) of different glial cells. However, this activation is followed by cell-type-specific transcriptional changes at later stages (5 dpi).

## Inhibition of the Cxcr3 and Tlr1/2 signaling pathways does not alter oligodendrocyte proliferation

Interference with the Cxcr3 and Tlr1/2 signaling pathways after brain injury did not result in the emergence of new cell types or states at either 3 or 5 dpi (Supplementary Fig. 11a). Nevertheless, the inhibition of the above-mentioned pathways elicited an overall downregulation of various inflammation-associated genes in the reactive glial clusters AG7, MG4, and OPCs2, particularly at 3 dpi (Fig. 6f, Supplementary Table 5). Furthermore, inhibition of the Cxcr3 and Tlr1/2 pathways after injury in the zebrafish telencephalon modulated oligodendrocyte proliferation, thereby decreasing oligodendrocytes in the injury vicinity[53]. To investigate the relevance of Cxcr3 and Tlr1/2 signaling in the mammalian context, we first sought to examine the cluster distribution of oligodendrocyte lineage cells among all conditions (INT, CTRL, and INH) and time points (3 and 5 dpi) (Supplementary Fig. 13a, b). We detected a shift in the cell distribution of the reactive OPC cluster OPCs2 between CTRL and INH samples most prominently at 3 dpi (Supplementary Fig. 13b). Intriguingly, the cell proportion that was shifted within OPCs2 in response to the inhibitor treatment represented mostly proliferating cells (Supplementary Fig. 13a). This indicated that, similar to our findings in zebrafish, OPC proliferation in response to SWI might be mediated by Cxcr3 and Tlr1/2 signaling. Therefore, we determined the proliferation ability of OLIG2$^+$ cells between inhibitor-treated and control animals by labeling all cells in S-phase with the DNA-base analog EdU (0.05 mg/g 5-ethinyl-2′-deoxyuridine i.p. injection 1 h before sacrifice) (Supplementary Fig. 13c). However, we could not observe changes in the number of proliferating oligodendrocytes (OLIG2$^+$ and EdU$^+$) or the number of OLIG2$^+$ cells near the injury site (Supplementary Fig. 13d–k).

In summary, the inhibition of Cxcr3 and Tlr1/2 signaling pathways after SWI in the mouse cerebral cortex, in contrast to findings in the zebrafish brain, did not alter oligodendrocyte proliferation or affected the overall number of oligodendrocyte lineage cells near the injury site early after injury, but did alter their inflammatory signatures.

## The Cxcr3 and Tlr1/2 signaling pathways regulate microglial activation in response to injury

The expression of inflammatory genes in microglia is tightly associated with their activation state[7,57]. Therefore, we assessed whether the downregulation of inflammatory genes induced by Cxcr3 and Tlr1/2 inhibition (Fig. 6f) might alleviate microglial reactivity. Hence, we examined the cell distribution of subclustered microglia among all three conditions (INT, CTRL, and INH) and time points (3 and 5 dpi)

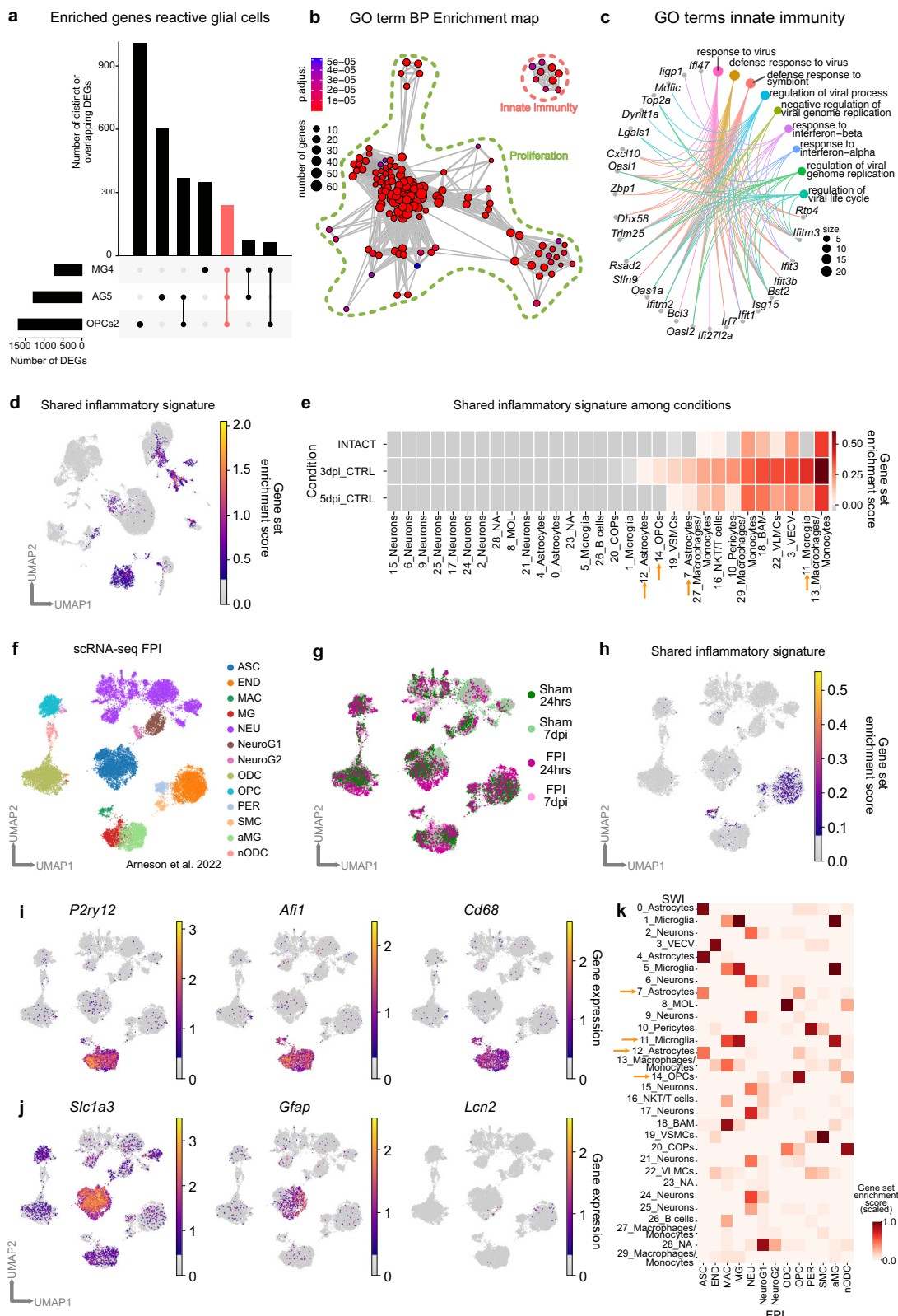

**a** Enriched genes reactive glial cells
**b** GO term BP Enrichment map
**c** GO terms innate immunity
**d** Shared inflammatory signature
**e** Shared inflammatory signature among conditions
**f** scRNA-seq FPI
**g**
**h** Shared inflammatory signature
**i** *P2ry12* *Afi1* *Cd68*
**j** *Slc1a3* *Gfap* *Lcn2*
**k**

Arneson et al. 2022

(Fig. 7a, b). As previously depicted in Fig. 4g, cells derived from intact samples were confined to the homeostatic clusters, whereas cells from the injured samples were distributed primarily in the reactive clusters at 3 dpi, and a transition toward the homeostatic clusters was noticeable at 5 dpi. A direct comparison of CTRL and INH samples indicated differences in the cell distributions, with a higher proportion of cells localized in the homeostatic clusters after Cxcr3 and Tlr1/2 inhibition (Fig. 7b). Although the discrepancy between conditions was already detectable at 3 dpi, the shift was more pronounced at 5 dpi (Fig. 7b). To further elucidate whether the detected shift in microglia distribution after Cxcr3 and Tlr1/2 pathway inhibition was accompanied by changes in overall cell morphology, we assessed microglia cell characteristics

**Fig. 5 | Stab wound injury induces common transcriptional changes in reactive glial subpopulations characterizing a specific reactive state. a** UpSet plot depicting unique (single points) or overlapping (connected points) DEGs (pval < 0.05, log$_2$ fold change >1.6, method *t*-test overestimated_variance). Vertical bars indicate the number of unique or shared genes between clusters. **b** GO term network analysis of the 241 commonly shared genes, associating shared DEGs with the biological processes of proliferation (green dashed line) and innate immunity (orange dashed line). Significance determined by method GO enrichment analysis on gene sets. **c** Chord diagram illustrating innate immunity GO terms from panel (**b**) and the corresponding genes. **d** UMAP plot depicting localization of the gene set score related to the shared inflammatory signature shown in panel (**c**) among all cell clusters from Fig. 4b. **e** Heatmap depicting the shared inflammatory signature gene set score. Data shown in panels (**a**)–(**e**) are derived from *n* = 12 intact animals over 3 independent experiments (5 scRNAseq libraries), *n* = 3 3dpi animals over 1 experiment (2 scRNAseq libraries), and *n* = 9 5dpi animals over 3 independent experiments (3 scRNAseq libraries). **f** UMAP plot illustrating single cells distributed among 13 distinct clusters in the FPI dataset. Clusters are annotated according to

Arneson et al.[50]. **g** UMAP plot depicting the distribution of cells isolated from Sham (green) and FPI (purple) cerebral cortices at 24 h or 7 dpi. **h** UMAP plot depicting the localization of the gene set score related to the shared inflammatory signature from panel (**c**). **i, j** UMAPs highlighting expression of example marker genes to identify reactive microglial (**i**) and astrocytic clusters (**j**). **k** Heatmap displaying cluster similarity of SWI (y-axis) and FPI datasets (x-axis). Orange arrows highlighting clusters of interest. DEGs differentially expressed genes, GO gene ontology, BP biological processes, UMAP uniform manifold approximation and projection, CTRL stab wound-injured control animals, dpi days post-injury, COPs committed oligodendrocyte progenitors, MOL mature oligodendrocytes, VECV vascular endothelial cells (venous), VSMCs vascular smooth muscle cells, VLMCs vascular and leptomeningeal cells, BAM border-associated macrophages, NKT cells natural killer T cells, NA not available, ASC astrocytes, END endothelial, MAC macrophages, MG microglia, NEU neurons, NeuroG1 neurogenesis1, NeuroG2 neurogenesis2, ODC oligodendrocytes, OPC oligodendrocyte precursor, PER pericytes, SMC smooth muscle cells, aMG activated microglia, nODC new oligodendrocytes, hrs hours, FPI fluid percussion injury, SWI stab wound injury.

with the automated morphological analysis tool described by Heindl et al.[58]. Brain sections from CTRL and INH-treated animals were labeled with an anti-IBA1 antibody, and areas near the injury site were analyzed (Fig. 7c). In response to brain damage, microglia transition from surveilling, highly ramified cells to activated, ameboid cells[59]. To capture the morphophysiological heterogeneity of microglia in the injured brain, we analyzed selected morphological features (circularity, soma volume, branch length & volume, number of major branches and nodes per major branch) of single microglial cells in at least 5 experimental animals per condition (Fig. 7c–g, Supplementary Fig. 14a–e). Indeed, we observed that every feature displayed a wide range distribution with multiple peaks independently of the condition (Fig. 7d–g, Supplementary Fig. 14d, e). Therefore, we first fitted these distributions with multiple peak function and compared the cumulative fits of each feature between the two experimental conditions. Microglia from INH-treated animals displayed significantly smaller cell somata (Fig. 7e) and appeared less compact (Fig. 7d) than microglia from CTRL animals. The inhibition of Cxcr3 and Tlr1/2 signaling pathways decreased the branch volume (Fig. 7f) without altering the total number of major branches and the branch length (Supplementary Fig. 14d, e). In addition, microglia from INH-treated animals appeared to be more ramified than CTRL microglia, because more nodes per major branch were detected (Fig. 7g). To assess if the changes in morphology following inhibitor treatment correlate with the expression of proteins used to assess the reactivity states of microglia, we compared the mean pixel intensities of the homeostatic marker P2Y12 (Fig. 7h–j) and the reactive markers IBA1 (Supplementary Fig. 14f–h) and CD68 (Supplementary Fig 14j–l) in the injury surrounding of inhibitor-treated and control animals at 5 dpi. We could not detect any changes between the experimental groups in all three markers analyzed (Fig. 7j, Supplementary Fig. 14h, l). Importantly, this correlated with no detectable changes in the mRNA levels based on the scRNA-seq analysis (Fig. 7k, Supplementary Fig. 14i, m).

In summary, our scRNA-seq data implied that Cxcr3 and Tlr1/2 pathway inhibition accelerates the transition from a reactive to a homeostatic microglial cell state early after injury. These findings were further supported by pronounced morphological changes in inhibitor-treated microglia, which are characteristic features of less reactive cells.

## Altered astrocyte response after Cxcr3 and Tlr1/2 pathway inhibition

To address the effects of Cxcr3 and Tlr1/2 pathway inhibition on astrocytes after brain injury, we subclustered astrocytes (Fig. 8a) and investigated the cell distribution among all conditions and time points (Fig. 8b). Astrocytes originating from intact conditions were evenly distributed among all homeostatic clusters. However, cells from stab-wounded brains were initially localized in both homeostatic and

reactive clusters at 3 dpi, whereas at 5 dpi, most cells were distributed among the reactive clusters. Comparison of astrocyte cell distribution of CTRL and INH samples indicated noticeable differences at 5 dpi. Most cells originating from the CTRL condition were distributed among the reactive clusters AG5, AG6, and AG7, whereas cells originating from the INH condition were largely confined to the reactive cluster AG5 (Fig. 8b). Interestingly, cluster AG5 exhibited lower expression of reactivity markers, such as *Gfap* and *Lcn2*, than the reactive clusters AG6 and AG7 (Supplementary Fig. 15a). In line with the shifted distribution of INH cells to cluster AG5, inhibitor-treated astrocytes also displayed lower expression of *Gfap* and *Lcn2* at 5 dpi (Supplementary Fig. 15b). To determine whether astrocyte reactivity was altered overall, we generated astrocyte reactivity scores (based on Hasel et al.[38]) and compared the reactivity gene set scores among INT, CTRL, and INH samples (Supplementary Fig. 15c). Generally, both reactivity scores (Cl4 and Cl8 in Supplementary Fig. 15c) were relatively lower in INH-treated samples at both time points (3 and 5 dpi). However, the fraction of astrocytes expressing these distinct gene sets was unchanged. Therefore, our analysis implied that the inhibitor treatment decreased astrocyte reactivity overall but was not sufficient to promote the return of reactive astrocytes to full homeostasis. In line with our findings from the scRNA-seq analysis, also by immunohistochemical assessment, we did not observe differences in reactive astrocyte states between stab-wounded control and inhibitor-treated mice at 5 dpi (Supplementary Fig. 15d–p). Both experimental groups showed comparable GFAP$^+$ cell accumulation (Supplementary Fig. 15d–l), NGAL (protein of *Lcn2*) intensities (Supplementary Fig. 15m–p), and numbers of NGAL$^+$ and GFAP$^+$ positive astrocytes in the injury vicinity (Supplementary Fig. 15e–l).

Furthermore, cluster AG5 was devoid of proliferating cells, because most cycling cells were confined to clusters AG6 and AG7, as indicated by the scRNA-seq proliferation score (Supplementary Fig. 15q, Supplementary Table 4). Interestingly, on the basis of our scRNA-seq analysis, interference with Cxcr3 and Tlr1/2 signaling after SWI decreased the fraction of proliferating astrocytes at 3 dpi, in line with the abundance of INH cells in cluster AG5 (Fig. 8b, Supplementary Fig. 15r).

To further investigate potential alterations in proliferation after inhibitor treatment, we assessed astrocyte proliferation with immunohistochemistry against GFAP (to identify reactive astrocytes) in combination with the DNA-base analog EdU (0.05 mg/g i.p. injection 1 h before sacrifice) at 3 dpi (Fig. 8c). Indeed, inhibition of the Cxcr3 and Tlr1/2 pathways after injury significantly decreased the proportion of proliferating (GFAP$^+$ and EdU$^+$) astrocytes in the injury vicinity (Fig. 8d–j). However, the total number of EdU$^+$ cells was not altered (Fig. 8d, g, k). In summary, our scRNA-seq analysis demonstrated decreased astrocyte reactivity and proliferation rates after inhibitor treatment. However, Cxcr3 and Tlr1/2 pathway inhibition, despite

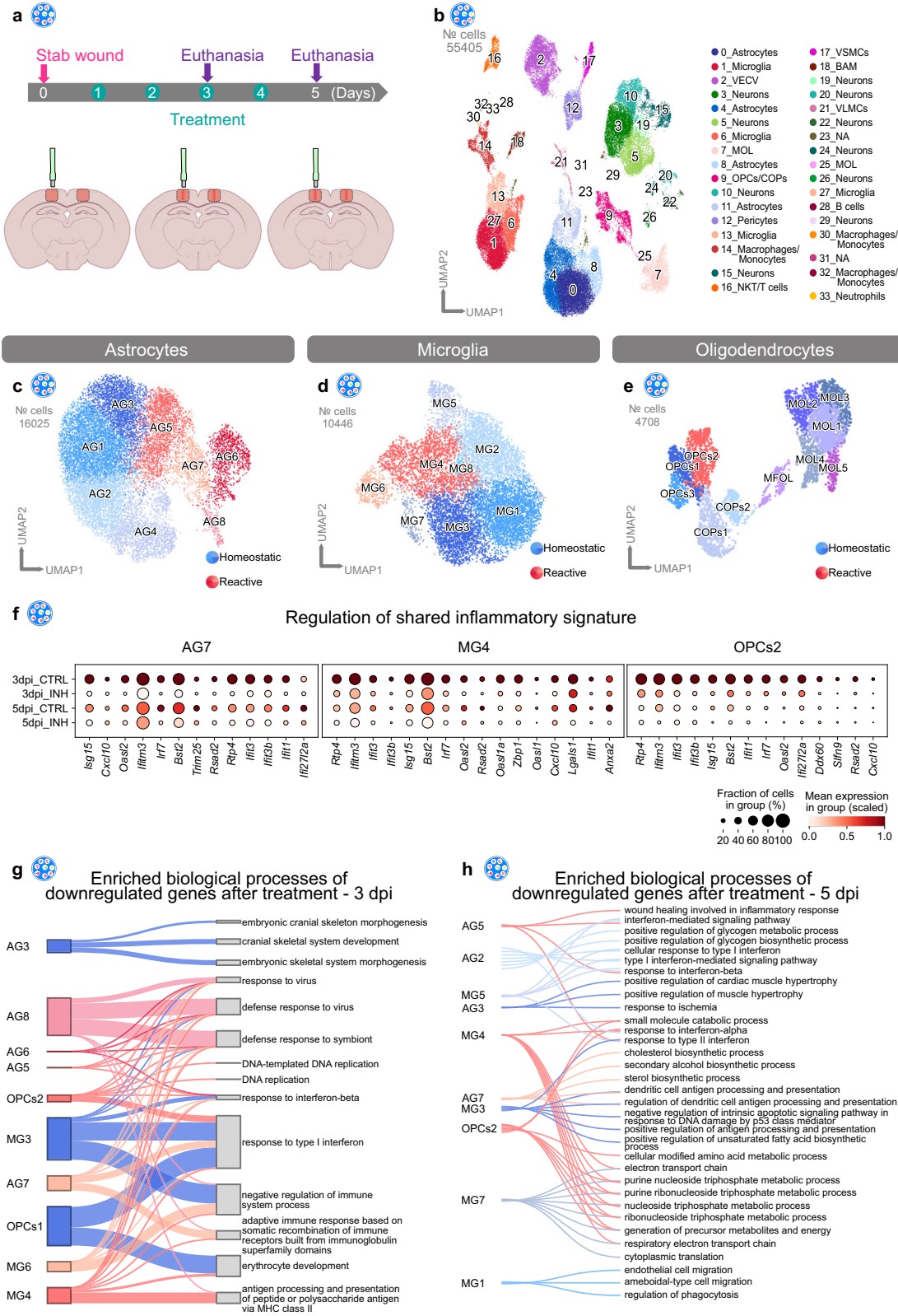

being sufficient to decrease astrocyte proliferation in vivo, did not completely revert astrocytes to homeostasis.

## Discussion

TBIs have complex pathophysiology involving responses of various types of cells[3,4]. However, most studies have focused on the responses of specific cell types, whereas few have extensively evaluated the interplay among these cells[24,27,60]. Therefore, we developed a comprehensive dataset profiling the transcriptional changes across various cell types in spatial and temporal contexts. Our study used the SWI model in mice[27,32], a mild injury model involving breakdown of the BBB and the activation of both glial and immune cells[27].

Spatial transcriptomic analysis of the stab-wounded cortex at 3 dpi revealed an injury-specific cluster, cluster VI, characterized by gene

**Fig. 6 | The Cxcr3 and Tlr1/2 pathways orchestrate the innate immune response shared among reactive glial cells. a** Experimental scheme for single-cell RNA-sequencing of intact, stab wound-injured control (3/5 dpi_CTRL) and stab wound-injured inhibitor-treated (3/5 dpi_INH) cerebral cortices with the 10x Chromium platform. Red masked areas on brain schemes indicate biopsy areas used for the analysis. **b** UMAP embedding of integrated and batch-corrected single-cell transcriptomes of 55405 cells. Cells were distributed among 34 distinct clusters, color-coded, and annotated according to their transcriptional identities. **c**–**e** UMAPs illustrating subclustering of astrocytes (8 clusters) (**c**), microglia (8 clusters) (**d**), and oligodendrocytes (11 clusters) (**e**). Cells were further assigned to homeostatic (blue), or reactive (red) clusters according to cell origin (Supplementary Fig. 11). **f** Dot plots depict decreased expression of various shared inflammatory genes from Fig. 5c in the reactive glial clusters AG7, MG4, and OPCs2 after inhibitor treatment. **g**, **h** Shankey diagram of GO terms linked to glial subclusters illustrating common and unique downregulated biological processes in response to Cxcr3 and Tlr1/2 pathway inhibition at 3 dpi (**g**) and 5 dpi (**h**). Only the top 3 biological process GO terms (based on adjusted *p*-value) are illustrated for each glial subcluster and the entire list of GO terms is shown in Supplementary Data 7. Line width indicates the adjusted *p*-value. Significance was determined by performing GO enrichment analysis on gene sets. Data shown in this figure are derived from $n = 12$ intact animals over 3 independent experiments (5 scRNAseq libraries), $n = 3$ 3dpi CTRL animals over 1 experiment (2 scRNAseq libraries), $n = 3$ 3dpi INH animals over 1 experiment (2 scRNAseq libraries), $n = 9$ 5dpi CTRL animals over 3 independent experiments (3 scRNAseq libraries) and $n = 6$ 5dpi INH animals over 2 independent experiments (3 scRNAseq libraries). UMAP uniform manifold approximation and projection, dpi days post-injury, OPCs oligodendrocyte progenitor cells, COPs committed oligodendrocyte progenitors, MOL mature oligodendrocytes, VECV vascular endothelial cells (venous), VSMCs vascular smooth muscle cells, VLMCs vascular and leptomeningeal cells, BAM border-associated macrophages, NKT cells natural killer T cells, NA not available, CTRL stab wound-injured control animals, INH stab wound-injured inhibitor-treated animals.

signature associated with various cell types. Cluster VI-enriched genes are involved in the regulation of processes associated with the immune system, in addition to angiogenesis and phagocytosis. Clearing dead cells and debris as well as re-establishing vasculature to ensure sufficient oxygen supply are critical defense mechanisms that occur early after brain damage[33], involving different types of glial and endothelial cells. Moreover, TBI induces immediate hyperexcitability of neurons at the infarct site that persists up to two months in animal models[35,61]. This hyperexcitability is correlating with the very early activation of IL13 signaling in neurons with neuroprotective function[62] implying that, in addition to glial cells, transcriptional changes in neurons could be one of the regulatory mechanisms restricting CNS damage. Indeed, the activity mediated expression of immediate early genes, specifically around the injury site has been associated with improved outcome after traumatic brain injury[63].

Despite the numerous advantages offered by recent techniques, the main hurdle in studying cellular interactions is to accurately identify the relevant cellular populations that shape the injured environment. While available stRNA-seq methods retain spatial information, they either lack the cellular resolution required to identify single cells[64] or are restricted to a subset of the transcriptome, introducing bias in data interpretation[65]. Conversely, several studies based on the scRNA-seq failed to associate the identified clusters to the injury because spatial information about clusters was lost[66]. Therefore, the combination of stRNA- with scRNA-seq represents an elegant approach to identify relevant regulatory cellular clusters and unveil transcriptomic changes coordinating the cellular response at the injury site. This is particularly important for focal pathologies, as the responses of astrocytes[17], OPCs[11,64], and microglia[64,67] are dependent on their proximity to the pathological site. In agreement with these data, our spatial gradient analysis revealed gene expression changes along pre-defined gradients extending from the injury core to the periphery, uncovering diverse descending patterns among injury-enriched genes. These data are well compatible with the idea that only some cellular clusters might contribute to the injury environment by either changing their transcriptome or/and their abundance around the injury site. Integration of scRNA-seq and stRNA-seq datasets allowed us to detect multiple cellular populations contributing to the injury-induced transcriptional profile. Specifically, probabilistic mapping of scRNA-seq clusters across the descending spatial gradient revealed that specific glial clusters were localized at the injury site. Importantly, all these clusters were detected mainly after SWI and expressed genes indicative of reactive glial cells. This is in line with emergence of new transcriptional states represented by new, reactive clusters enriched at the injury site that define a new environment of the injury-specific spatial cluster VI.

The emergence of reactive glial clusters showed distinct patterns of reactivity. Microglial clusters displayed a uniform response to injury as all microglial clusters were enriched at the injury site, with cluster 11_Microglia exhibiting the highest enrichment score. Importantly, these reactive microglia are distinct from the specific activation patterns (disease-associated microglia signature) observed in the APP model of neurodegeneration[64], suggesting pathology-specific responses of microglia. Contrary to microglia, astrocytes exhibited heterogeneous responses upon SWI. Clusters 12_Astrocytes and 23_Astrocytes accumulated around the injury site, however the remaining astrocytic clusters were underrepresented in the injury area compared to the rest of the cortex. Interestingly, the injury-enriched astrocytic clusters 12_Astrocytes and 23_Astrocytes displayed unique features corresponding to their location and gene signatures. Cluster 12_Astrocytes for example, expressed high levels of *Gfap*, whereas cluster 23_Astrocytes might represent the recently described atypical astrocytes, which, after focal brain injury, rapidly downregulate GFAP among other astrocytic proteins[31]. We also observed cluster 15_OPCs as the only OPC cluster showing enrichment at the injury core. Interestingly, we did not identify specific reactive OPC hallmarks within this cluster despite clear evidence of reactive OPCs at the injury site[8,68]. This result may be partly explained by the unknown signature of reactive OPCs, because only an increase in proliferation and expression of CSPG4 have been used to identify reactive OPCs to date[9]. Interestingly, compared to glial clusters, neuronal clusters were not locally associated with the injury-induced cluster, suggesting a minor role in the overall contribution to the injury-specific cluster VI.

Considering that the integration of scRNA-seq and stRNA-seq datasets indicated the presence of distinct glial cell subtypes similarly contributing to the injury environment, we sought to determine if these glial cells exhibit any specific transcriptional features linked to their reactivity state. The addition of the 5 dpi dataset allowed us to analyze temporal changes in response to injury. In-depth analysis of astrocytes, microglia, and oligodendrocytes uncovered distinctive features of injury-responsive glia. We identified genes associated with proliferation and the activation of innate immune processes that are commonly shared by all three reactive glia populations. Indeed, proliferation of microglia[69], astrocytes[36,70], and OPCs[8,71] is typically evoked by brain injury. However, proliferation cannot be considered as a hallmark of reactive glia as it is not observed across all CNS pathologies[72]. Consequently, we investigated whether the shared inflammatory signature could be a common feature of glial cells. Indeed, the genes defining the shared inflammatory signature, as identified in our study, are also expressed in reactive microglia and astrocytes induced by MCAO[51]. Notably, in the MCAO dataset OPCs were not captured thus the expression of genes defining the shared inflammatory signature in reactive OPCs cannot be assessed. In addition, the genes belonging to the shared inflammatory signature are partially present in LPS-induced reactive astrocytes[38]. However, not all shared inflammatory genes were expressed in reactive glia. This finding implies a common expression of

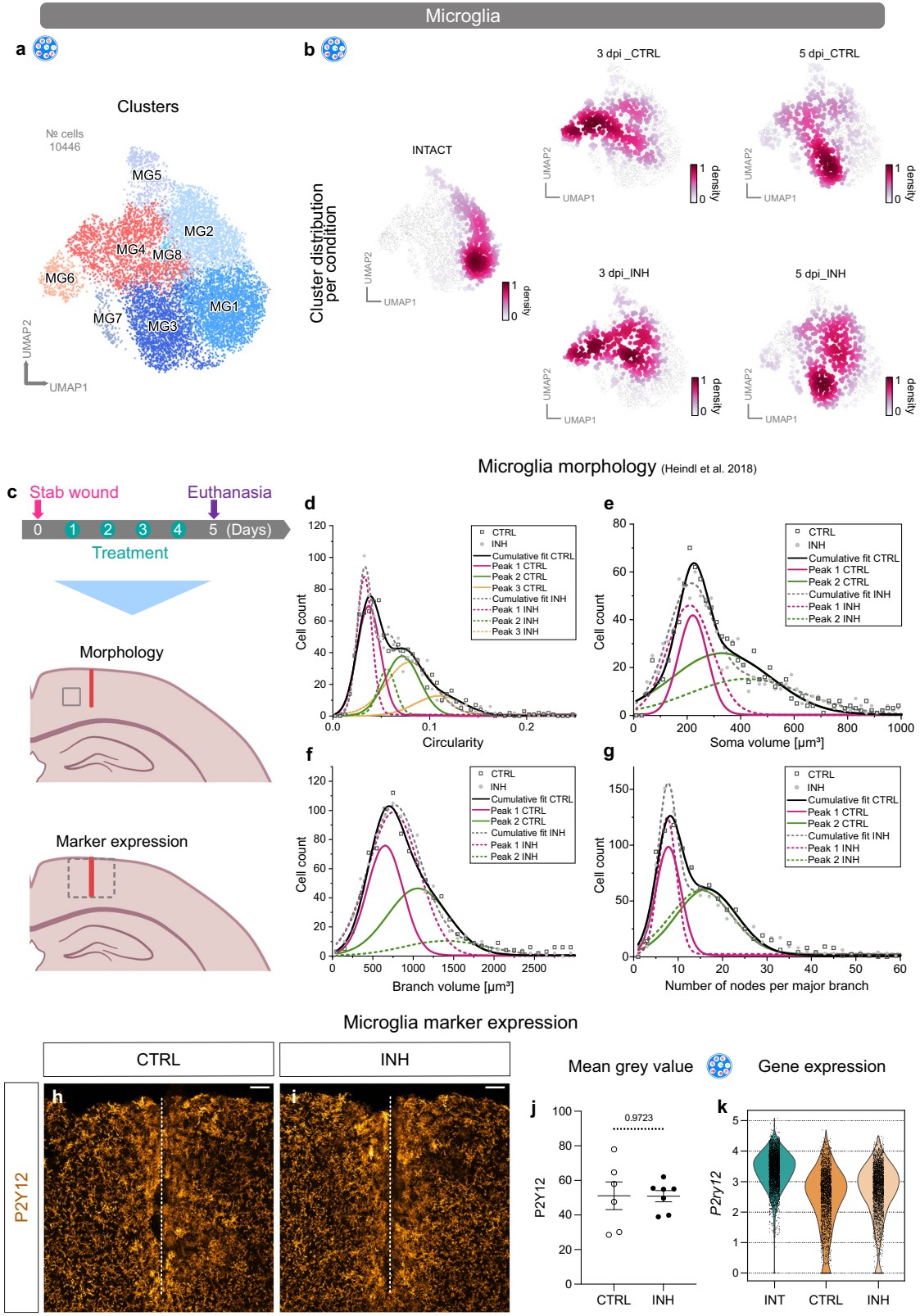

core innate immunity-associated genes in different cell types in response to a variety of stimuli. However, each pathological condition further triggers distinct processes, which are uniquely coordinated in each pathology. In line with this, the FPI injury model[50] does not trigger glia-wide expression of genes defining the shared glial response and these gene signature was expressed only in few microglial cells. The absence of the shared inflammatory signature in astrocytes and OPCs

could be explained by either distinct regulatory mechanisms controlling glial reactivity following this type of TBI or different reactivity level of glial cells. Indeed, we observed a prominent expression of the shared inflammatory signature in glial cells 3 dpi after SWI, which was gradually diminished by 5 dpi. In line with this finding, we could not identify classical glial reactivity markers such as *Lcn2* in the focal FPI dataset, suggesting that in this dataset the glial cells are not at the maximal

**Fig. 7 | Microglial reactivity following Cxcr3 and Tlr1/2 pathway inhibition.**
**a**, **b** UMAPs illustrating subclusters of microglia (**a**) and cell distributions (**b**) among those subclusters at all time points (intact, 3 dpi, and 5 dpi) and conditions (CTRL and INH). Data shown in (**b**) are downsampled to an equal number of cells between time points and conditions. Data shown in panels (**a**) and (**b**) are derived from $n = 12$ intact animals over 3 independent experiments (5 scRNAseq libraries), $n = 3$ 3dpi CTRL animals over 1 experiment (2 scRNAseq libraries), $n = 3$ 3dpi INH animals over 1 experiment (2 scRNAseq libraries), $n = 9$ 5dpi CTRL animals over 3 independent experiments (3 scRNAseq libraries) and $n = 6$ 5dpi INH animals over 2 independent experiments (3 scRNAseq libraries). **c** Experimental paradigm to assess microglial reactivity. Gray boxes on the mouse brain schemes highlight the analyzed areas. Red lines indicate the injury cores. **d**–**g** Plots depicting the distribution of microglial circularity (**d**), soma volume (**e**), branch volume (**f**), and number of nodes per major branch (**g**) of $n_{CTRL} = 6$ and $n_{INH} = 5$ animals. Source data are provided as Source Data file. Data are fitted with multiple peak functions. The fit parameters are depicted in Supplementary Data 8 for each fit. **h**, **i** Representative images of P2Y12 staining in CTRL (**h**) and INH-treated (**i**) mice. Dashed white lines indicate the injury core. All images are full z-projections of confocal z-stacks. **j** Dot plot depicting the mean gray value of P2Y12$^+$ signal in the injury vicinity of CTRL and INH-treated mice at 5 dpi. Data are shown as mean ± standard error of the mean. Each data point represents one animal. Source data are provided as Source Data file. Statistics have been derived from $n_{CTRL} = 6$ and $n_{INH} = 7$ animals. *p*-values were determined with unpaired *t*-test (two-tailed). **k** Violin plots depicting expression levels of *P2ry12* in microglia (scRNA-seq) in intact, 5 dpi CTRL and 5 dpi INH condition. Data are derived derived from $n = 12$ intact animals over 3 independent experiments (5 scRNAseq libraries), $n = 9$ 5dpi CTRL animals over 3 independent experiments (3 scRNAseq libraries) and $n = 6$ 5dpi INH animals over 2 independent experiments (3 scRNAseq libraries). Scale bars: **h**, **i**: 50 μm. UMAP uniform manifold approximation and projection, dpi days post-injury, CTRL stab wound-injured control animals, INH stab wound-injured inhibitor-treated animals, INT intact.

reactivity level compared to the reactivity level in the SWI model. These findings, therefore, suggest that the shared inflammatory signature could define a specific feature of reactive glia that reflects their strong reactivity state[36].

The inflammatory signature present in the reactive clusters MG4, AG5, and OPCs2 included several genes belonging to IFN-I pathway, a pathway associated with strong neuroinflammatory response[73]. Among these genes, interferon regulatory factor 7 (*Irf7*), a transcription factor crucial for IFN-I activity[74], and *Cxcl10*, encoding a well-characterized ligand of the Cxcr3 pathway[49], were detected. Previous studies have demonstrated that IRF7 induces type I IFNs through the activation of *Tlr2*, thus resulting in the transcription of several mediators, including CXCL10[75,76]. In addition, the Tlr2/Irf7 signaling axis has been associated with microglia-mediated inflammation after subarachnoid hemorrhage in mice[77]. Furthermore, we have recently demonstrated that Cxcr3 and Tlr1/2 regulate OPC accumulation at injury site in the zebrafish brain in a redundant and synergistic manner[53]. Our data support the concept that the same innate immune pathways are responsible for initiating the response in injury-induced glial reactivity. In line with this concept, the systemic inhibition of the Cxcr3 and Tlr1/2 pathways modulates the shared inflammatory signature in glial cells and induces similar alterations across different glial cells shortly after injury. However, these initial changes are diversified at 5 dpi inducing cell-type-specific alterations in different glial cells. This interpretation is in line with the above-discussed discrepancy to the focal FPI dataset.

Our data suggest that the Cxcr3 and Tlr1/2 pathways regulate only a subset of features defining each reactive glia population. For instance, we observed a decrease in the number of proliferating astrocytes proximal to the injury site without an alternation in the immunoreactivity of typical reactive astrocytic markers such as GFAP or NGAL after inhibition of the Tlr1/2 and Cxcr3 pathways. Similarly, following systemic inhibition of the Tlr1/2 and Cxcr3 pathways, microglia in the injury vicinity exhibited morphological features reminiscent of less reactive cells, as compared to control animals. However, no changes were detected in the level of IBA1 or P2Y12, typical markers indicating the microglial state[78]. In line with these observations, our transcriptome analysis demonstrated that both astrocytic and microglia populations isolated from Tlr1/2 and Cxcr3 deficient SWI animals, as opposed to the glial cells isolated from control SWI animals, show a shift towards the homeostatic clusters. This shift indicates higher similarities to cells isolated from the intact brains, further emphasizing the significance of our unbiased approach in addressing glial reactivity.

In conclusion, the present study provides a comprehensive transcriptomic dataset for analyzing early events after TBI with respect to changes in time, space, and cell type. Additionally, this dataset provides an excellent platform to examine the interplay of a variety of cells in response to injury. A better understanding of injury pathophysiology may provide more opportunities for developing new therapeutic strategies.

## Methods
### Animals
All surgeries were performed on 8–12 week old male mice (*Mus musculus*), housed, and handled under the German and European guidelines for the use of animals for research purposes. Room temperature was maintained within the range of 20–22 °C, while the relative humidity ranged between 45–55%. The light cycle was adjusted to 12 h light:12 h dark period. Room air was exchanged 11 times per hour and filtered with HEPA-systems. All mice were housed in individually ventilated cages (2–5 individuals per cage) under specified-pathogen-free conditions. All animals had free access to water and food. The cages were equipped with nesting material, a red corner house, and a rodent play tunnel. Soiled bedding was removed every 7 days. Given the previous reported sexual dimorphism in glial cell responses to pathologies[57], we decided to exclusively focus our study on males to ensure data consistency and enable a targeted analysis within our pre-defined research scope. C57Bl6/J animals were purchased from Charles River (strain #000664). NG2-CreERT$^2$ breeding animals (Cspg4$^{tm1.1(cre/ERT2)Fki}$, MGI:5566862, Huang et al.[79]) were kindly provided by the Kirchhoff laboratory and crossed to the commercially purchased CAG-GFP reporter mouse line (Jackson Laboratory, strain #024636). NG2-EYFP breeding pairs (Cspg4$^{tm1.1Trot}$, MGI: 3846720, Karram et al.[80]) were kindly obtained from the University of Mainz. NG2-EYFP animals were kept on the C57Bl6/N and NG2-CreERT$^2$ animals on the C57Bl6/J background. Experiments were approved by the government of Upper Bavaria (animal license number: ROB-55.2-2532.Vet_02-20-158) and Saarland state's "Landesamt für Verbraucherschutz" in Saarbrücken/Germany (animal license number: 17/2023). Anesthetized animals received a stab wound lesion in the cerebral cortex by inserting a thin knife (19G, Alcon #8065911901) into the gray matter using the following coordinates from Bregma: RC: −1.2; ML: 1–1.2 and from Dura: DV: −0.6 mm. To produce stab lesions, the knife was moved over 1 mm back and forth along the anteroposterior axis from −1.2 to −2.2 mm as described before[32]. Animals were euthanized 3 and 5 days after the injury (dpi). Cervical dislocation was used for all sc- and stRNA-seq experiments, while transcardial perfusion was conducted for histochemical experiments (for more details see section "Tissue preparation").

For the induction of Cre-mediated recombination in NG2-CreERT$^2$xCAG-GFP mice, tamoxifen (40 mg/ml, Sigma #T5648) was administered orally (20G, Merck #CAD9921). Animals received tamoxifen every second day (400 mg/kg) for a total of 3 times. Mice were injured two weeks after the last tamoxifen administration and sacrificed at 5 dpi by transcardial perfusion.

For the treatment experiments, animals received an inhibitor cocktail mix composed of the Cxcr3 antagonist NBI 74330 (100 mg/kg, R&D Systems #4528) and the Tlr1/2 inhibitor CU CPT 22 (3 mg/kg, R&D

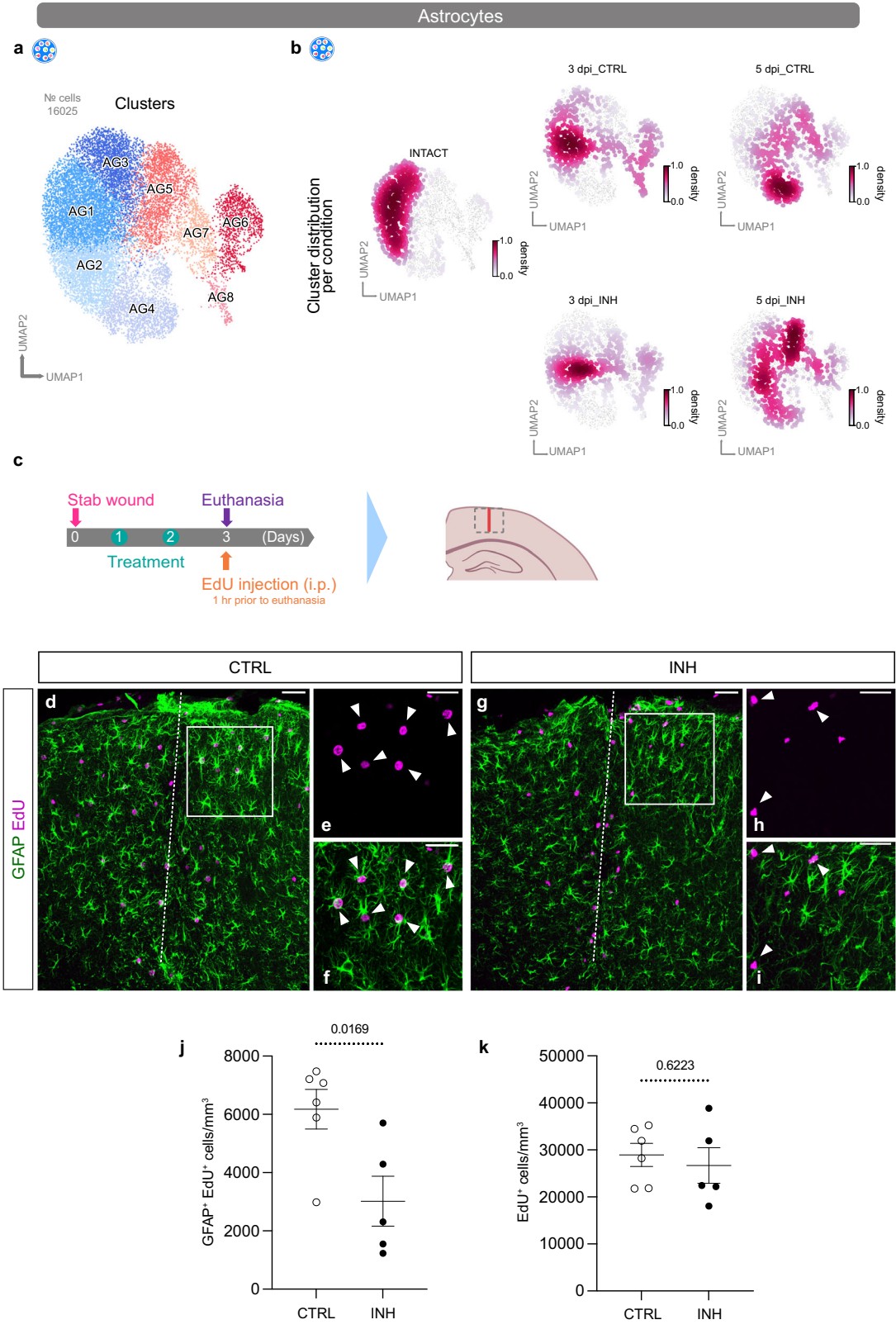

Systems #4884) by gavage feeding (20G, Merck #CAD9921). Both compounds were individually dissolved in DMSO (Sigma-Aldrich #D2438), combined, and further diluted in corn oil (Sigma-Aldrich #C8267). The vehicle solution consisted of DMSO diluted in corn oil and was administered to all control animals. Animals received the first dose of the inhibitor cocktail mix or the vehicle solution 24 h after stab

wound injury and daily thereafter. NBI 74330 and CU CPT 22 dosages were obtained from previous publications[81,82].

To analyze the proliferative capacity of glial cells we injected 5-Ethinyl-2′-deoxyuridine (EdU, 0.05 mg/g, ThermoFisher #E10187) intraperitoneally and animals were sacrificed by transcardial perfusion 1 h after injection.

**Fig. 8 | Proliferation of reactive astrocytes is decreased after Cxcr3 and Tlr1/2 pathway inhibition. a, b** UMAPs illustrating subclusters of astrocytes (**a**) and cell distributions (**b**) among those subclusters at all time points (intact, 3 dpi, and 5 dpi) and conditions (CTRL and INH). Data in (**b**) are downsampled to an equal number of cells between time points and conditions. Data are derived from $n = 12$ intact animals over 3 independent experiments (5 scRNAseq libraries), $n = 3$ 3dpi CTRL animals over 1 experiment (2 scRNAseq libraries), $n = 3$ 3dpi INH animals over 1 experiment (2 scRNAseq libraries), $n = 9$ 5dpi CTRL animals over 3 independent experiments (3 scRNAseq libraries) and $n = 6$ 5dpi INH animals over 2 independent experiments (3 scRNAseq libraries). **c** Experimental paradigm for assessing astrocyte proliferation. The dashed gray box on mouse brain scheme indicates the analyzed area. The red line indicates the injury core. **d, e** Representative overview images of proliferating GFAP+ (green) and EdU+ (magenta) astrocytes in CTRL (**d**)

and INH-treated (**e**) animals. White dashed lines highlight injury cores. Micrographs (**e, f, h, i**) are magnifications of white boxed areas in (**d**) and (**g**), respectively. White arrowheads in micrographs indicate colocalization of EdU (**e, h**) with GFAP+ astrocytes (**f, i**). All images are full z-projections of confocal z-stacks. **j, k** Dot plots depicting density of proliferating (GFAP+ and EdU+) astrocytes (**j**) and total density of proliferating (EdU+) cells (**k**) in CTRL and INH-treated animals. Data are shown as mean ± standard error of the mean. Each data point represents one animal. Source data are provided as Source Data file. Statistics in (**j**) and (**k**) have been derived from $n_{CTRL} = 6$ and $n_{INH} = 5$ animals. *p*-values were determined with unpaired *t*-test (two-tailed). Scale bars: **d, g**: 50 µm, **e, i**: 20 µm. UMAP uniform manifold approximation and projection, dpi days post-injury, EdU 5-ethynyl-2′-deoxyuridine, i.p. intraperitoneal injection, CTRL stab wound-injured control animals, INH stab wound-injured inhibitor-treated animals.

## Tissue preparation

Mice were deeply anesthetized and transcardially perfused with phosphate-buffered saline (PBS) followed by 4% paraformaldehyde (PFA) (wt/vol) dissolved in PBS. Brains were postfixed in 4% PFA overnight at 4 °C, washed with PBS and cryoprotected in 30% sucrose (Carl Roth #4621.2) at 4 °C. Mouse brains used to assess microglia morphology were embedded in 3% agarose and cut coronally at 100 µm thickness using a vibratome (HM 650V, Microm). Otherwise, brains were embedded in frozen section medium Neg-50 (Epredia #6502), frozen, and subsequently sectioned using a cryostat (Thermo Scientific CryoStar NX50). Coronal sections were collected either at a thickness of 20 µm on slides for RNAscope (Epredia #J1800AMNZ) or 40 µm for free-floating immunohistochemistry.

## Immunohistochemistry

For immunohistochemistry, sections were blocked and permeabilized with 10% normal goat serum (NGS, vol/vol, Biozol #S-1000)/donkey serum (NDS, vol/vol, Sigma-Aldrich #566460) and 0.5% Triton X-100 (vol/vol, Sigma-Aldrich #T9284), dissolved in 1xPBS while being incubated overnight at 4 °C with the corresponding primary antibodies. Following primary antibodies were used: anti-CD68 (rat 1:600, BioRad, MCA1957T), anti-GFP (chick 1:400, Aves Labs, GFP-1020), anti-GFAP (mouse 1:500, Sigma-Aldrich, G3893), anti-IBA1 (rabbit 1:500, Wako, 019-19741), anti-NGAL (recognizing product of Lcn2 gene; rabbit 1:500, ThermoFisher, PA5-79590), anti-OLIG2 (mouse 1:200, Millipore, MABN50), anti-P2Y12 (rabbit 1:750, AnaSpec, AS-55043A), anti-SERPINA3N (goat 1:500, R&D Systems AF4709-SP). Sections were washed with PBS and incubated with secondary antibodies dissolved in 1×PBS solution containing 0.5% Triton X-100 for 2 h at room temperature. Following secondary antibodies were used: donkey anti-chick IgY A488 (1:1000, Dianova 703-545-155), donkey anti-goat IgG A647 (1:1000, Jackson Immuno Research 705-605-003), goat anti-mouse IgG1 A546 (1:1000, ThermoFisher A-21123), goat anti-mouse IgG2a A488 (1:1000, ThermoFisher A-21131), goat anti-mouse IgG1 A488 (1:1000, Thermo-Fisher A-21121), goat anti-rabbit IgG A546 (1:1000, ThermoFisher A-11010), goat anti-rabbit IgG A633 (1:1000, ThermoFisher A-21070), goat anti-rat IgG A488 (1:1000, ThermoFisher A-11006), goat anti-rat A546 (1:1000, ThermoFisher A-11081). For nuclear labeling, sections were incubated with DAPI (final concentration of 4 µg/mL, Sigma #D9542) for 10 min at room temperature. EdU incorporation was detected by Click-iT™ EdU Alexa Fluor™ 647 Imaging Kit (ThermoFisher Scientific #C10340) according to the manufacturer's instructions. Staining procedure for microglia morphology analysis was performed as described by Heindl et al.[58]. Briefly, sections were blocked for 1 h with goat serum blocking buffer containing 2% normal goat serum (NGS vol/vol, Biozol #S-1000), 1% bovine serum albumin (BSA, Sigma-Aldrich #A2153), 0.1% cold fish skin gelatin (Sigma-Aldrich #G7765), 0.1% Triton X-100 (vol/vol, Sigma-Aldrich #T9284), and 0.05% Tween20 (Sigma-Aldrich #P7949) dissolved in 1xPBS. Sections were incubated overnight with an IBA1 antibody (rabbit 1:200, Wako, 019-19741) dissolved in primary

antibody buffer containing 1% BSA (Sigma-Aldrich #A2153), 0.1% cold fish skin gelatin (Sigma-Aldrich #G7765), 0.1% Triton X-100 (vol/vol, Sigma-Aldrich #T9284), and 0.05% Tween20 (Sigma-Aldrich #P7949) dissolved in 1× PBS. Sections were washed with PBS and incubated with the secondary antibody (goat anti-rabbit IgG A546 1:500, ThermoFisher A-11010) for 2 h. For nuclear labeling, sections were incubated with DAPI (final concentration of 4 µg/mL, Sigma #D9542) for 10 min at room temperature. Stained sections were mounted on glass slides (Epredia #AG00000112E01MNZ10) with Aqua-Poly/Mount (Polysciences #18606).

## In situ hybridization

RNA in situ hybridization was performed using RNAscope® Multiplex Fluorescent Reagent Kit (ACD, 323110) according to the manufacturer's instructions. Briefly, brain sections were fixed in 4% paraformaldehyde at 4 °C for 15 min, ethanol-dehydrated (Carl Roth #9065.4), treated with $H_2O_2$ (ACD, 322381) and protease-permeabilized for 20 min at 40 °C. Brain sections were then incubated for 2 h at 40 °C using the following probes: *Ifi27l2a* (ACD, 88617), *Serpina3n* (ACD, 430191-C2), *Lcn2* (ACD, 313971-C3), *Cd68* (ACD, 316611-C2), *Oasl2* (ACD, 534501), and *Cxcl10* (ACD, 408921-C3). Signal was amplified according to the manufacturer's instructions (User manual Cat.Nr: 320293, Fluorophore Opal 520: Akoya Biosciences FP1488001KT). Subsequently, sections were processed with immuno-histochemistry analysis as described above. The primary antibodies used in combination with RNAscope® were as follows: chick antibody to GFP (1:400, Aves Lab, GFP-1020), goat antibody to GFAP (1:300, Abcam, ab53554), rabbit antibody to IBA1 (1:500, Wako, 019-19741).

## Image acquisition and processing

Confocal microscopy was performed at the core facility bioimaging of the Biomedical Center (BMC) with an inverted Leica SP8 microscope using the LASX software (Leica). Overview images were acquired with a 10x/0.30 objective, higher magnification pictures with a 20×/0.75, 40×/1.30 or 63×/1.40 objective, respectively. Image processing was performed using the NIH ImageJ software (version 2.1.0/1.53f). To acquire overview images, single images were stitched using the ImageJ plug-in tool pairwise stitching[83]. Images utilized for the microglia morphology analysis were acquired with an 40×/1.30 objective with an image matrix of 1024 × 1024 pixel, a pixel scaling of 0.2 µm × 0.2 µm and a depth of 8-bit as previously published[58]. Microglia morphology parameters were extracted by using the publicly deposited MATLAB script of Heindl et al.[58] [https://github.com/isdneuroimaging/mmqt]. For the morphological microglia analysis, selected microglial features (circularity, soma volume, branch volume, number of nodes per major branch, branch number, branch length) were extracted from each animal ($n_{CTRL} = 6$, $n_{INH} = 5$ animals). Furthermore, microglia parameters for each animal were grouped according to their condition ($n_{CTRL} = 923$, $n_{INH} = 922$ cells) and further analyzed. Downsampling of control microglia was conducted using the tool Data Sampler in Orange (v.3.32.0). The distribution of cells according to extracted

parameters was fitted using the multipeak function (Supplementary Data 8) without the definition of number of peaks. The multipeak function with highest chi-square value was used for the distribution comparison. The distributions were compared using the Microcal Origin package (version 2021b OriginLab Corporation, Northampton, MA, USA).

## Spatial transcriptomics analysis

Mouse brains from 3 dpi ($n = 1$) or intact ($n = 1$) C57Bl6/J mice were embedded and snap frozen in an isopentane (Carl Roth #6752.5) and liquid nitrogen bath as recommended by 10x Genomics (Protocol: CG000240). During cryosectioning (Thermo Scientific CryoStar NX50) brains were resected to generate a smaller sample (Fig. 1a) and two 10 μm thick coronal sections of the dorsal telencephalic regions were collected in one capture area. The tissue was stained using H&E staining (Protocol: CG000160) and imaged with the Carl Zeiss Axio Imager.M2m Microscope using a 10x objective. The libraries were prepared with the Visium Spatial Gene Expression Reagent Kits (10x Genomics PN-1000187, protocol: CG000239) with 18 min permeabilization time and sequenced on an Illumina HiSeq1500 instrument and a paired-end flowcell (High output) according to manufacturer protocol, with sequencing depth of 55231 (Intact) and 75398 (3 dpi) mean reads per spot. Sequencing was performed in the Laboratory for Functional Genome Analysis (LAFUGA).

Data were mapped against the mouse reference genome mm10 (GENCODE vM23/Ensembl 98; builds versions 1.2.0 and 2020A from 10x Genomics) with Space Ranger 1.2.2. Both datasets were analyzed, and quality checked following the Scanpy[84] and Squidpy[85] pipeline, selecting spots with at least 1500 reads and a maximum 45% mitochondrial fraction. Normalization and log-transformation was performed using the counts per million (CPM) strategy with a target count depth of 10,000 using Scanpy's[84] normalize_total() and log1p functions. Following cell count normalization and scaling (function scale in Scanpy), experimental groups were integrated. Highly variable gene (HVGs) selection was performed via the function highly_variable_genes() using the Cell Ranger flavor with default parametrization, obtaining 2000 HVGs. Unsupervised clustering of cells was done using the Leiden algorithm[86] as implemented in Scanpy. This allowed classification of multiple clusters based on marker genes selected using test_overestim_var() between the normalized counts of each marker gene in a cluster against all others (function rank_genes_groups in Scanpy). The layer marker score was performed using the function score_genes (as implemented in Scanpy) based on established marker genes (Supplementary Table 2) described by Zeisel et al.[34]. Gene ontology enrichment analysis was performed using the function enrichGO() (R package: clusterProfiler[56]) on the marker genes for cluster VI (indicated above) selecting the genes with pval<0.05 and $\log_2 fc > 1$ and the top 10 functions of the three aspects (MF: Molecular Function; CC: Cellular Component; BP: Biological Process) and were presented on a dot plot.

## Single-cell analysis

The lesioned gray matter of the somatosensory cortex of C57BL/6J mice at 3 dpi and 5 dpi or the corresponding region of the intact cortex were isolated using a biopsy punch (∅ 0.25 cm, Plano #15111-25), and the cortical cells were dissociated at a single-cell level using the Papain Dissociation System (Worthington #LK003153) followed by the Dead Cell Removal kit (Miltenyi Biotec #130-090-101), according to manufacturer's instructions. Incubation with the dissociating enzyme was performed for 60 min. For each dissociation 3 experimental animals were used, with 2 biopsy punches collected per animal.

Single-cell suspensions were resuspended in 1× PBS with 0.04% BSA (10× DPBS: gibco #14200-067, UltraPure™ BSA (50 mg/mL): Invitrogen AM2616) and processed using the Single-Cell 3′ Reagent Kits v2 or v3.1 (PN-120267 or PN-1000269, Supplementary Table 6) from 10x

Genomics according to the manufacturer instructions. In brief, this included generation of single-cell gel beads in emulsion (GEMs), post-GEM-RT cleanup, cDNA amplification, and library construction. Illumina sequencing libraries were sequenced on a HiSeq 4000 or NovaSeq6000 system (with an average read depth of 30,000 raw reads per cell) according to the manufacturer's instructions for each version. Sequencing was performed in the genome analysis center of the Helmholtz Center Munich.

Transcriptome alignment of single-cell data was done using Cell Ranger v3.0.2 and v6.0.0 against the mouse reference genome mm10 (GENCODE vM23/Ensembl 98; builds versions 1.2.0 and 2020 A from 10x Genomics). Quality Control (QC) of mapped cells from all datasets integrated was done using recommendations by Luecken and Theis[87] selecting cells with at least 1000 genes, maximum of 50,000 reads, and 25% mitochondrial fraction. Doublets were removed using the Scrublet framework[88], clusters expressing multiple lineage genes were identified as mixed populations and were removed from further analysis. Normalization was performed using the scran[84] package (R package) followed by log-transformation using Scanpy's log1p functions[84]. Highly variable gene (HVGs) selection was performed via the function highly_variable_genes using the Cell Ranger flavor with default parametrization, obtaining 2000 HVGs. Following cell count normalization and scaling, (function scale in Scanpy) experimental groups were batch corrected with scVI[89,90]. Unsupervised clustering of cells was done using the Leiden algorithm[86] as implemented in Scanpy. This allowed classification of multiple main clusters based on marker genes selected using test_overestim_var between the normalized counts of each marker gene in a cluster against all others (function rank_genes_groups in Scanpy). The top 50 marker genes were used for the cluster annotation using the online available databases for the mouse brain (http://mousebrain.org/adolescent/genesearch.html) and the immune cells (http://rstats.immgen.org/MyGeneSet_New/index.html). Additionally, we generated gene expression scores using the function score_genes (as implemented in Scanpy) based on established marker genes (Supplementary Table 3) of the main cell populations in the adult mouse brain to further confirm the cluster annotation. Visualization of cell groups is done using Uniform Manifold Approximation and Projection (UMAP)[91], as implemented in Scanpy. Differential gene expression analysis between treated and control conditions was performed using the tool diffxpy (https://diffxpy.readthedocs.io/en/latest/index.html) using the Wald test. Of note, since some glial subclusters are comprised of only few cells, the differential gene analysis was not performed in these subclusters.

All the comparisons of the overlapping genes were performed using the R package UpSetR[92] which provides an efficient way to visualize the intersecting gene set in an UpSet plot. For cluster comparison additionally, the gene ontology analysis was performed using the R package clusterProfile[56], using the functions compareCluster() (fun:enrichGO) or enrichGO(). The visualization of the functional enrichment results was done using the following visualization methods from the R package enrichplot[56]: enrichment map (function: emaplot()) (based on the pairwise similarities of the enriched terms calculated by the pairwise_termsim() function); and the Gene-Concept Network plot (function: cnetplot()). In addition, the R package networkD3 function sankeyNetwork() was used to visualize the top 3 GO terms for each clusters between CTRL and INH treatment at 3 and 5 dpi.

For the reanalysis of the MCAO scRNA-seq dataset[51] we obtained the processed files from Gene Expression Omnibus (GEO) with the accession number GSE174574 and followed the same quality control (QC), normalization, batch correction, and annotation procedures as listed above. For the reanalysis of the FPI scRNA-seq dataset[50] we obtained the processed files for the frontal cortex samples (Sham and TBI, 24 h and 7 dpi) from GEO with accession number GSE180862 and followed the same normalization and batch correction, as listed above. The clusters were annotations based on Arneson et al.[50]. For

calculating the shared inflammatory signature in both datasets, we used the genes depicted in Fig. 5c. Similarity matrix was calculated based on the top 20 genes defining each cluster of the SWI dataset.

### Spatial alignment of the scRNA-seq data

For the spatial localization of the scRNA-seq data, we used the Python package Tangram[48], focusing on the 3 dpi control condition and using only the cortical cluster of the Visium dataset in order to have the same anatomical region. We selected the training genes using the tool AutoGeneS[93] and used 439 training genes as the union of the top informative marker genes of each cluster in the scRNA-seq data that were detected in the Visium profiles. To find the spatial alignment for the scRNA-seq we used the Tangram[48] function map_cells_to_space() which gave us the probabilistic mapping score. Additionally, we segmented the H&E image, using the Squidpy[85] function segment which was used for deconvolving the Visium data using the Tangram[48] functions count_cell_annotations() and deconvolve_cell_annotations().

### Spatial gradient analysis

Spatial gradients extending from the lesion core towards perilesional regions were defined using SPATA[94] and its successor SPATA2[41] (under development; https://themilolab.github.io/SPATA2/). The injury core is defined as the dense area at the injury site based on the H&E staining. The perilesional area covers 500 μm of adjacent region as this has been reported to contain tissue with reactive gliosis[11,17,36]. The Scanpy-processed object described above was used as input. Both lesion cores were manually annotated based on the H&E staining using createImageAnnotations(). Visium spots were binned into concentric circles using the following arguments: n_bins_circle = 13, binwidth = "95 μm". Spots from non-cortical clusters (III,V,X,XIII,XIV,XV,XII,XVI) were excluded from the analysis using the argument bcsp_exclude. Genes with >50 total counts were screened for their correlation with pre-defined gradients (e.g. linear descending) using imageAnnotation-Screening(), for both injuries separately. Spot metadata derived from Scanpy and Tangram, as well as genes that correlated most strongly with selected pre-defined gradients (sorted by p_value_mean) were plotted using plotIasHeatmap_merge() and plotIasRidgeplot_merge(), custom adaptations of original SPATA2 functions, in which values represent the bin-wise mean of both injuries. Descending models included 'linear_descending', 'immediate_descending', 'abrupt_descending', 'late_descending'. Ascending models included 'linear_ascending', 'immediate_ascending', 'abrupt_ascending', 'late_ascending' (see function showModels()). For screening of gene sets, the following sets were downloaded from MsigDB (https://www.gsea-msigdb.org/gsea/msigdb/mouse/collections.jsp): Biocarta, KEGG, Reactome, WikiPathways, GO (MF/CC/BP), Hallmark. Per gene set, the mean expression of all included genes was calculated and screened for correlation with the same pre-defined gradients as described for single genes. A snapshot of the utilized state of SPATA2 including custom functions is available at https://github.com/simonmfr/SPATA2/tree/publicationCK.

### Statistics and reproducibility

Mice were selected randomly and allocated into different experimental groups. No additional randomization was used during data collection. Investigators were blinded while acquiring and analyzing data from different experimental groups. Each data point in every dot plot represents one animal. For all quantifications depicted in Figs. 7j, 8f, g, and Supplementary Figs. 13j, k, 14h, l, 15k, l, o a minimum of two sections per animal were analyzed. In each section, an area of 300 μm (150 μm on each side of the injury) was selected and either the pixel covered area or the number of positive cells in all individual z-planes of an optical z-stack (step size 1.41 μm) was quantified. For the quantifications of the shared inflammatory genes (Supplementary Fig. 8c–t) three sections per animal were analyzed. In each section, two to three images were taken around the injury core and the number of positive cells in all individual z-planes

of an optical z-stack (step size 0.3422 μm) was quantified. Additionally, to account for variations in section thickness, total cell numbers were normalized to the section depth. Pixel coverage and intensity comparisons were carried out using a defined number of individual z-planes. Statistical analysis was performed using GraphPad Prism (version 9.3.1) and OriginPro (version 2021b OriginLab Corporation, Northampton, MA, USA). The normality of the data distribution was assessed using the Shapiro–Wilk test. Normally distributed data were tested for significance with unpaired t-test (two-sided). The data distributions of the microglia morphology features depicted in Fig. 7d–g and Supplementary Fig. 14d, e were compared using Compare Datasets: fitcmpdata function in Origin. All statistical tests used are specified in the corresponding legends. No statistical methods were used to pre-determine sample sizes. stRNA-seq experiment as well as experiments to visualize the injury-induced cluster VI (Fig. 1g, h) was performed once. scRNA-seq experiments were performed in at least two replicates. Quantification of astrocyte (Fig. 8j, k) and oligodendrocyte proliferation (Supplementary Fig. 13j, k) in control- and inhibitor-treated animals at 3 dpi was repeated twice with similar results. Experiments to assess morphological microglia features (Fig. 7d–g, Supplementary Fig. 14d, e) were conducted in two independent experimental rounds with reproducible results. Astrocyte reactivity at 5 dpi was performed once (GFAP coverage (Supplementary Fig. 15k), GFAP/NGAL+ density (Supplementary Fig. 15l)) or twice (NGAL intensity (Supplementary Fig. 15o)). Experiments to determine microglia reactivity at 5 dpi (Fig. 7j, Supplementary Fig. 14h, l) was performed twice with similar results. Visualization of shared inflammatory genes in Supplementary Fig. 8c–t was performed twice with reproducible results. Data points in Fig. 7j, Supplementary Fig. 14h, l and 15o were obtained from NG2-CreER$^{T2}$xCAG-GFP (4× CTRL + 4× INH) and C57Bl6/J (2× CTRL + 3× INH) animals. Data points did not cluster according to the different mouse lines, suggesting that NG2-CreER$^{T2}$xCAG-GFP animals display a similar glial reaction to SWI as C57Bl6/J animals. The data points in Supplementary Fig. 15k and l were obtained exclusively from NG2-CreER$^{T2}$xCAG-GFP animals. Data points in Supplementary Fig. 8b originated from NG2-EYFP mice and in Fig. 8j, k and Supplementary Fig. 13j, k from C57Bl6/J animals.

### Reporting summary

Further information on research design is available in the Nature Portfolio Reporting Summary linked to this article.

## Data availability

All sequencing data generated in association with this study are available in the Gene Expression Omnibus as a SuperSeries under accession number GSE226211. scRNA-seq data are available under accession number GSE226207 and spatial transcriptomic under GSE226208. The mouse reference genome mm10 [https://www.10xgenomics.com/support/software/cell-ranger/downloads/cr-ref-build-steps] was used for the data alignment. The online available databases for the mouse brain [http://mousebrain.org/adolescent/genesearch.html] and the immune cells [http://rstats.immgen.org/MyGeneSet_New/index.html] were used for the cluster annotation of the scRNA-seq data. scRNA-seq data from other studies referenced in Fig. 5 and Supplementary Fig. 9 are available from the GEO with the accession numbers GSE180862 and GSE174574, respectively. The plots in Supplementary Fig. 9f-g were generated from the online searchable database GliaSeq containing the scRNA-seq analysis by Hasel et al.[38] [https://liddelowlab.shinyapps.io/GliaSeqPro/]. Source data are provided with this paper.

## Code availability

Details of the analysis pipeline libraries are listed in Methods and available at https://github.com/NinkovicLab/Koupourtidou-Schwarz-et-al.[95] for scRNA-seq stRNA-seq and https://github.com/simonmfr/SPATA2/tree/publicationCK, for the spatial gradient analysis. Microglia

morphology parameters were extracted using the publicly deposited MATLAB script of Heindl et al.[58] [https://github.com/isdneuroimaging/mmqt].

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

## Acknowledgements

We are particularly grateful to the entire Neurogenesis and Regeneration group members for their experimental inputs and discussions and Dr. Alessandro Zambusi for critical reading of the manuscript. We would especially like to thank Dr. Steffanie Heindl for her help with the microglia 3D reconstruction. We acknowledge the support of the following core facilities: the Bioimaging Core Facility and Bioinformatic Core Facility at the BioMedical Center of LMU Munich, the Laboratory for Functional Genome Analysis (LAFUGA), and the Sequencing Facility at the Helmholtz Zentrum Munchen. This work was supported by the German research foundation (DFG) by the SFB 870 (JN, MG); TRR274/2 2024-40885537 (INST 186/1543-2; JN, MG); GO 640/19-1 (JN, MG); SPP 1738 "Emerging roles of non-coding RNAs in nervous system development, plasticity & disease" (JN), SPP1757 "Glial heterogeneity" (JN); DSPP2191 "Molecular mechanisms of functional phase separation" (DO 1804/4-2)(JN),the Excellence Strategy within the framework of the Munich Cluster for Systems Neurology (EXC 2145/1010 SyNergy – ID 390857198)(JN, MG, MD), Fritz Thyssen Stiftung (JN) and Ampro Helmholtz Alliance (JN), FOR2879/2 (no. 405358801, MG), the SPP 2306 Ferroptosis (project no. 461629173, MG); German Research Foundation (DFG) as part of the Munich Cluster for Systems Neurology (EXC 2145 SyNergy – ID 390857198, MD), the CRC 1123 (B3; MD), DI 722/16-1 (ID: 428668490/40535880, MD), DI 722/13-1, and DI 722/21-1 (MD); a grant from the Leducq Foundation (MD); the European Union's Horizon Europe (European Innovation Council) programme under grant agreement No 101115381 (MD); ERA-NET Neuron (MatriSVDs, MD), and the Vascular Dementia Research Foundation (MD).

## Author contributions

C.K., V.S., and J.N. conceived the project and experiments. C.K., V.S., J.F.S., T.S.E., R.B., S.S., and X.B. performed experiments, with C.K. and V.S. conducting subsequent data analysis. C.K., H.A., and S.F. performed the bioinformatic analyses. C.K., V.S., and J.N. wrote the manuscript with input from all authors. J.N., M.G., M.D., F.K., and F.J.T. supervised research, designed experiments, and acquired funding.

## Funding

## Competing interests

The authors declare the following competing interests: F.T. consults for Immunai Inc., Singularity Bio B.V., CytoReason Ltd, and Omniscope Ltd, and has ownership interest in Dermagnostix GmbH and Cellarity. All other authors declare no competing interests.
