## [Peer Review File · Nature Communications]

Shared inflammatory glial cell signature after stab wound injury, revealed by spatial, temporal, and cell-type-specific profiling of the murine cerebral cortexREVIEWER COMMENTS

Reviewer #1 (Remarks to the Author):

In this study Ninkovic and colleagues sought to characterize the cell-type specific responses and transcriptional signatures of mouse brain after injuries. They utilized a TBI model by creating stab wounds in mouse cortex, then performed spatial transcriptomics and single-cell RNA sequencing of the brain tissue sections at multiple time points post injury. From the resulting data, they identified specific cell populations representing altered cell states for astrocytes, microglia and OPCs, mapped these cells to the tissue neighborhood surrounding the wound sites, and further identified injury-related genes and pathways. This led to the identification of inflammatory programs across multiple glia cell types (but not neurons). They then focused on two key genes (Cxcr3 and TIR1/2) that were upregulated in these active cells, by treating injured animals with chemical inhibitors. They observed some level of reduction in terms of inflammatory response, validating their hypothesis derived from the transcriptomics data. Overall, this is a rigorously designed study aiming to address important questions related to traumatic brain injury. They used cutting edge experimental tools and computational methods. The experiments using Cxcr3 and TIR1/2 inhibitors represent a nice validation. The overall quality is high, and the findings are timely and significant to the community of brain injury studies. I think it clearly meet the standards of Nature Communication.

I do have two questions here: (1) the majority of TBI cases in human are due to concussions. How representative a stab-wound injury model is to those concussion led TBIs? (2) Is there any differences on the animal behaviors or brain functions, for the mice 3-day or 5-day post injury, with or without NBI74330 and CU CPT 23 treatments? I'm not asking new experiments to address these questions, but perhaps the authors can say something in the Discussions. I also found the current Discussions section a bit lengthy and repeating some points already covered in the previous sections. I would recommend to trim this section and keep only key messages.

A few other minor points:

Figure 3a: add "Days" like in Figure 4a

Figure 3c: add legends for the cyan and red colors.

Figure 3f: is it possible to add a scale bar?

Figure 7f: list the sample size N

NBI74330 and CU CPT 23 treatments: how was the dosage determined?

Reviewer #2 (Remarks to the Author):

This paper by Koupourtidou et al. describes the transcriptomic response of glial cells to traumatic brain injury in the neocortex of adult mice. A combined spatial and single cell transcriptome approach has been applied to identify the specific responses and states of typical glial cells. A main focus was on identifying shared injury-related genes between the glial populations. The paper describes an important resource for future investigations on the impact of TBI in the mammalian brain.

The paper describes how the transcriptomics findings are in line with known responses of glial cells to brain injury: a first and swift response of microglia, followed by oligodendrocytes, and then astrocytes; and also a quick return to homeostasis for the MG, and a slower one for the astrocytes. From this perspective the data do not appear as innovative. The big plus lies in opening up the large datasets to the broader research community, as a resource.

At times the paper is a bit boring to read, the authors could have done a better job in linking their data to for example previous literature that also describes the impact of distance to the lesion, for example in electrophysiology or in immediate early gene studies, in the context of brain lesions or

stroke. There are many papers describing a change in response in relation to distance to the lesion both for cortical cells and molecules, in homonymous zones in the other hemisphere and so on. These earlier studies focused more on neurons, but really deserve to be integrated in the discussion.

Specific comments

Results

Lines 103-104: the description of the position of the coronal sections on the glass slides for spatial transcriptome analysis reads weird – why 'parenchymal' sections? This phrasing is never used for mouse tissue; please call it grey matter.

Lines 117-119: why would you have expected layers of the cortex to be changed upon TBI? Again, previous research has ruled out such an effect (see above).

Lines 148-149: what exactly is different for the CD68 distribution compared to Icn2 or serpin3n? the word 'defined' is not saying a lot.

Line 155: Why 13 concentric circles? Of what size? How defined? Size wise: distance in um? In figure 2, none of the panels is informative about size; how was it decided to collect spatial data up to 1mm from the lesion? How are core and perilesion area determined? Please indicate what is perilesion in these figures.

Panels B and D are quite hard to read. Is it possible to sort the data per pattern and then to explain the different patterns more logic? Is interpretation of genes of similar pattern informative?

Line 205-206: please verify reference to figure panels: correct? Should 2c be 3b?

Lines 327-334: please rewrite, this part about the comparison to a dataset of human iPSC derived astrocytes is very hard to follow.

The high number of abbreviations reduces the readability of the paper, especially the part about the inhibitors (from line 427 onwards).

Discussion:

Lines 599 and onwards: the comparison to earlier zebrafish work would benefit from rewriting. What is the mean message here?

Line 643: is it also possible that the double inhibitor approach prevents the switch from homeostatic to reactive microglia instead of accelerating the opposite?

Line 651: A coordinated response of astrocytes? What does this mean, please clarify.

Figures:

Figure 3: panel e: it would have been nice to see how well the cell types may cluster based on color pattern; for example, cluster 3 astrocytes and BAMs ? Was this analysed?

Figure 5: panels e-g: the cxcl10 signal in panel g is not convincing; please provide a better illustration.

Figure 6: panels g, h: these panels are hard to read or interpret; the legend is also not helping a lot: either rework or remove; as is there is no point in keeping them in the main figures.

Figure 8 panels d,e and ED Fig 13: So from around 25k Edu+ cells about 6000 are oligos and 2500 astrocytes? What other cells are Edu Positive? MG? I am puzzled about the high percentage of Edu in Astrocytes (25 %!), this is in adult cortex; Is this only in the vicinity of the injury? Exactly when in time after the lesion? How much time was there for Edu incorporation? This finding deserves

better discussion.

ED fig 1 panel d: the white dotted line for the lesion is hard to see; panel f: see earlier comments on similar graphs. Please check legend as well since the reference to panels appears wrong: should e not be d?

ED fig3: the title 'Remaining' clusters can have two meanings: Do you mean 'Cells remaining in the lesion site/core'? Or 'all other cells'? I guess you mean the latter, please clarify, and apply to other figures and legends.

ED fig 6: I would have liked to see a panel for Aldh1l1.

ED fig 7: Please rephrase the legend so that the content and meaning of the figure becomes more clear.

ED fig 9: The figure is not really convincing, I also did not get it why here data for galectin-1 are shown; what is the plus?; Why is GFP in yellow in panel c? and IBA1 in blue?; Panel f: the clusters have no nr or name? to what other umap is this linked?

M&M:

Line 1119: 'operations' should be 'surgery'; please also add n to reveal nr of animals used in the study; and explain why only male mice are used;

Reviewer #3 (Remarks to the Author):

In this study, Koupourtidou and Schwarz et al. leveraged cutting-edge spatial transcriptomics approaches to provide comprehensive and unbiased spatial, temporal, and cell-type-specific profiling of gene expression in the murine cerebral cortex following a stab injury. From these studies, they uncovered an underappreciated injury-induced innate immunity-shared glial signature involving the Tlr1/2 and Cxcr3 signaling pathways. They next leveraged specific inhibitors/antagonists of Tlr1/2 and Cxcr3 in order to validate the functional relevance of these combined pathways in regulating glial responses in their stab model of traumatic brain injury (TBI). Here they found that combined abrogation of Tlr1/2 and Cxcr3 dampened the effects of stab injury on microglial morphology and astrocyte proliferation in the cerebral cortex. The research presented in this manuscript is novel, well-controlled, and presented in a manner that is both clear and provocative. While this is unquestionably a valuable resource for the TBI field, a few specific shortcomings and issues were noted, which I believe need to be addressed in order to solidify their central findings and conclusions.

Specific comments:

1. The TLR1/2 signaling axis has been most extensively studied in the context of bacterial infection models where it has been shown to be a major orchestrator of innate immunity in response to bacterial lipoproteins. It would be interesting for them to speculate briefly somewhere in the text as to what they believe is driving TLR1/2 signaling in their stab TBI model. Most notably, it would be particularly interesting to know if they suspect that an endogenously-derived damage-associated molecular pattern (DAMP) is driving TLR1/2 activation in their model or if they believe that the stab puncture is now exposing the brain to bacterial ligands present on the skin surrounding the wound.

2. While their microglial morphology studies presented in Figure 7 are a good starting point to evaluate the effects of combined Cxcr3 and Tlr1/2 inhibition on microglial activation, additional studies are needed to strengthen their conclusions here. To this end, they should evaluate additional markers of microglial activation at the protein level to bolster their data interpretation

here as changes in microglial morphology do not always provide an accurate quantification of microglial activation levels. Likewise, it would also help to boost the impact and significance of their spatial transcriptomics findings if they validated some of their major astrocyte findings in Figure 8 (e.g. changes in GFAP and LCN2 levels).

3. To increase the rigor of their findings presented in Figure 7f and Extend Data (ED) Figure 14a, they should provide average values per mouse as opposed to all acquisition data. Most notably, there appears to be data nesting in these figure panels (i.e. multiple data points from a low number of mice) and this could significantly skew and confound data interpretation.

4. Data quantification and statistical significance is needed to accompany the representative images presented in Figures 5 e-g and ED Fig 8 b-g.

Reviewer #4 (Remarks to the Author):

In this report the authors performed spatial and single cell transcriptomic analysis in the cortex of mice that received stab wound injury. They describe shared signatures of astrocytes microglia and oligodendrocytes at day 3 and 5 post injury.

They aim to report functional relevance of the shared signatures however, as they state they primarily confirmed previous findings from their zebrafish models and other findings on injury to the brain. The conclusions are overstated, and the results presented here do not support the conclusions; there are no functional outcome measures. The data reported do not bring significant novel information that advance the field.

The major limitations are listed below:

The transcriptomic data is both inconsistent and many details are left out. For the initial finding in Fig 3, it is a total of 6322 cells analyzed, then they subdivide this into 30 clusters. The number of cells per cluster is not given so we can't assess if this is even enough cells to make all of their additional conclusions (if you estimate an even split which obviously it is not you would say ~200 cells/cluster which is low, especially bc likely many of the clusters are less than this). They do not note how many animals are used for this analysis, given the low cell yield it seems like 1 per group which again is not sufficient to draw conclusions.

In Figure 4, now they are comparing intact, 3 dpi, and 5 dpi, but it looks like the 3 dpi from Fig 3 is used, which isn't all that unusual except, in Fig 3 there are 3646 cells for 3 dpi and now in Fig 4 there are 3637, it is not noted why this number changes. Again, they do not list how many mice are added for each condition. Additionally, they subcluster the microglia, astrocytes, and oligos, but again don't list the total number of these cells to determine if they are powered enough to subcluster. Given these limitations, I would be hesitant to believe any of their single cell analysis (including spatial).

They analyzed 16567 cells in intact brains, 3637 at 3dpi and 13658 at 5dpi, why there is 5 fold less cells at 3dpi? Same for the analysis with the treatments (Cx3Cr1 and Tlr2) they analyzed 16649 from intact animals and 3643 from 3dpi 4613 and 13766 and 15142 from 5dpi. This is a major caveat to the interpretation of the results.

In Figure 5 they compare their dataset with an LPS injection model, but not published TBI datasets? It is unclear why they would not compare with published TBI datasets. In the discussion they mentioned that the compared "the signature that are also present in other brain pathologies such as LPS" (line 585) . LPS is not a brain pathology.

For Fig 7 - it is not clear how many mice are analyzed per group. The figure legend notes 450 microglia per condition, but not the number of mice per condition. Additionally, Fig 8 has the number of mice graphed where as Fig 7 has the number of cells, why is there this inconsistency?

For Fig. 9 when they add in the antagonist, they need to clarify if they are adding both the Cxcr3 and Tlr1/2 antagonist together. Why they did not test adding them independently? Secondly, they confirm the chemical compounds in an OPC knock-out cell line, why not microglia or astrocyte? Again it looks like they are using the analysis from Fig 3 and 4, but now 3 dpi has 3643 cells (so different from Fig 3 and Fig 4). Additionally, the 3 dpi + INH only has 4613 cells, whereas 5 dpi has 15142 - why is there this large difference? Again the number of mice per group is not noted.

The injury used is a stab wound injury that does not recapitulate fully the chronic TBI consequences as they themselves state for instance in the resolution of some of the signatures by 5dpi. The manuscript is misleading as it referred to TBI while this is a very specific kind of injury and should be specified:

In the title (line 1), Results (Line 92, 242) As a result the intro should be tailored to this and not to all TBI.

They only use male mice (not clear how many) it is not justify why female mice were not included.

Line 58: Activated microglia should be changed with Reactive Microglia as it is well accepted that we can't describe microglia as "activated".

"Sacrificed" should always be replaced with Euthanized.

Response to REVIEWER COMMENTS

Reviewer #1 (Remarks to the Author):

In this study Ninkovic and colleagues sought to characterize the cell-type specific responses and transcriptional signatures of mouse brain after injuries. They utilized a TBI model by creating stab wounds in mouse cortex, then performed spatial transcriptomics and single-cell RNA sequencing of the brain tissue sections at multiple time points post injury. From the resulting data, they identified specific cell populations representing altered cell states for astrocytes, microglia and OPCs, mapped these cells to the tissue neighborhood surrounding the wound sites, and further identified injury-related genes and pathways. This led to the identification of inflammatory programs across multiple glia cell types (but not neurons). They then focused on two key genes (Cxcr3 and TIR1/2) that were upregulated in these active cells, by treating injured animals with chemical inhibitors. They observed some level of reduction in terms of inflammatory response, validating their hypothesis derived from the transcriptomics data. Overall, this is a rigorously designed study aiming to address important questions related to traumatic brain injury. They used cutting edge experimental tools and computational methods. The experiments using Cxcr3 and TIR1/2 inhibitors represent a nice validation. The overall quality is high, and the findings are timely and significant to the community of brain injury studies. I think it clearly meet the standards of Nature Communication.

We thank this reviewer for the constructive input. We believe that we can fully address reviewer concerns as follows:

Major concerns:

(1) the majority of TBI cases in human are due to concussions. How representative a stab-wound injury model is to those concussion led TBIs?

We find this comparison very difficult, as outcome of concussions is highly dependent on the severity of the impact. Each of the two models does have specific features (e.g. edema for concussion or direct damage of the blood vessels in stab wound). Nevertheless, the glial reactivity is observed in both models. Therefore, we compared our scRNAseq with scRNAseq coming from the concussion model in mice (fluid percussion injury). This analysis reveals an intriguing difference in the temporal dynamics of the glial cell reaction. The analysis is now presented in the Figure 5f-j. Moreover, we included the discussion on this aspect of glial reactivity in the revised manuscript.

(2) Is there any differences on the animal behaviors or brain functions, for the mice 3-day or 5-day post injury, with or without NBI74330 and CU CPT 23 treatments? I'm not asking new experiments to address these questions, but perhaps the authors can say something in the Discussions. I also found the current Discussions section a bit lengthy and repeating some points already covered in the previous sections. I would recommend to trim this section and keep only key messages.

We did not observe any obvious behavioral differences. However, we do not exclude any as we did not perform the full behavioral analysis. In the revised version, we provided more concise discussion that also included discussion on possible neuroprotective role of innate immunity pathways.

Minor points:

- (1) Figure 3a: add "Days" like in Figure 4a

The labelling is added to the revised figure.

- (2) Figure 3c: add legends for the cyan and red colors.

The labelling is added to the revised figure legend.

- (3) Figure 3f: is it possible to add a scale bar?

Yes, it is possible and scale bar is added to all images illustrating spatial transcriptomics.

- (4) Figure 7f: list the sample size N

Such dot-plots for the analysis of the microglial reactivity are standard in the field (Heindl et al. 2018, Benakis et al. 2022). We analyzed 6 control (total 923 cells) and 5 inhibitor (total 922 cells)-treated animals. These numbers are now stated in the revised material and method section.

- (5) NBI74330 and CU CPT 23 treatments: how was the dosage determined?

These inhibitors have been already used and we chose already used doses. This is now clearly stated in Method section including the respective citation.

Reviewer #2 (Remarks to the Author):

This paper by Koupourtidou et al. describes the transcriptomic response of glial cells to traumatic brain injury in the neocortex of adult mice. A combined spatial and single cell transcriptome approach has been applied to identify the specific responses and states of typical glial cells. A main focus was on identifying shared injury-related genes between the glial populations. The paper describes an important resource for future investigations on the impact of TBI in the mammalian brain.

The paper describes how the transcriptomics findings are in line with known responses of glial cells to brain injury: a first and swift response of microglia, followed by oligodendrocytes, and then astrocytes; and also a quick return to homeostasis for the MG, and a slower one for the astrocytes. From this perspective the data do not appear as innovative. The big plus lies in opening up the large datasets to the broader research community, as a resource. At times the paper is a bit boring to read, the authors could have done a better job in linking their data to for example previous literature that also describes the impact of distance to the lesion, for example in electrophysiology or in immediate early gene studies, in the context of brain lesions or stroke. There are many papers describing a change in response in relation to distance to the lesion both for cortical cells and molecules, in homonymous zones in the other hemisphere and so on. These earlier studies focused more on neurons, but really deserve to be integrated in the discussion.

It is indeed an interesting point. These reports are, however, largely linked to the very early response and could not be seen in our first time point (3 days after injury). Nevertheless, we included in the discussion the possibility that neuronal gene expression could have an important regulatory role that was missing in the initially submitted manuscript.

Specific comments

Results

(1) Lines 103-104: the description of the position of the coronal sections on the glass slides for spatial transcriptome analysis reads weird – why ‘parenchymal’ sections? This phrasing is never used for mouse tissue; please call it grey matter.

In our sections, we capture also a portion of white matter despite the fact that we do not injure it. Therefore, we feel that the term grey matter would not be the most appropriate. We do understand the concern of our reviewer that the parenchyma is not often used for the mouse tissue. Therefore, we would change the term “parenchymal section” with term “brain sections containing the dorsal telencephalic regions”.

(2) Lines 117-119: why would you have expected layers of the cortex to be changed upon TBI? Again, previous research has ruled out such an effect (see above).

We do not expect the alternation. However, it is important to see this in our analysis. We used this as the necessary control. We agree that the incorporation of references previously demonstrating no change in the neuronal layering would be necessary here and we added it in the revised manuscript.

(3) Lines 148-149: what exactly is different for the CD68 distribution compared to *lcn2* or *serpin3n*? the word 'defined' is not saying a lot.

The spread of the injury induced expression for those factors is different. While the CD68 is confined to the injury site, the *Lcn2* and *SerpinA3* are showing broader expression. This is now described in the revised manuscript.

(4) Line 155: Why 13 concentric circles? Of what size? How defined? Size wise: distance in μm ? In figure 2, none of the panels is informative about size; how was it decided to collect spatial data up to 1mm from the lesion? How are core and perilesion area determined? Please indicate what is perilesion in these figures.

We did not pre-define the number of circles. We used equally spaced circles around the injury to cover a radius of 500 micrometers (1 mm in total) excluding the subcortical regions. We decided to use 500 micrometers as this corresponds to the area with glial reactivity after stab wound injury that was previously reported. This is now stated in the method part of the revised manuscript. Each circle is 95 micrometer and 13 circles fitted on the section before reaching the midline of the injured hemisphere. The injury core is defined based on the H&E staining as H&E dense area and perilesional tissue is defined as surrounding tissue (up to 500 micrometers on both sides). This is also stated in the method part in the revised manuscript and labelled in the respective figure 2.

(5) Panels B and D are quite hard to read. Is it possible to sort the data per pattern and then to explain the different patterns more logic? Is interpretation of genes of similar pattern informative?

We now arranged clusters based on their enrichment along the spatial gradient. This indeed fits the expected localization of known gene expression. The revised panel is included and discussed in the respective result section of the revised manuscript.

(6) Line 205-206: please verify reference to figure panels: correct? Should 2c be 3b?

In line 205-206 we refer to the clusters that appear after injury. The immune cell clusters that appear after injury are expressing high levels of *Ccr2* and this is shown in ED Fig. 2c. In our view, we referred to the correct panel.

(7) Lines 327-334: please rewrite, this part about the comparison to a dataset of human iPSC derived astrocytes is very hard to follow.

We have now replaced this part with the comparison between stab wound injury (our data set) and fluid percussion injury (classical TBI model), as suggested by reviewer 4. We agree with the reviewer 4 that such comparison is more informative. Due to the space limitations, we decided to remove the comparison to the iPSC derived astrocytes.

(8) The high number of abbreviations reduces the readability of the paper, especially the part about the inhibitors (from line 427 onwards).

We simplified our abbreviations in the revised manuscript and use only control and inhibitors to define the data set.

Discussion:

(1) Lines 599 and onwards: the comparison to earlier zebrafish work would benefit from rewriting. What is the mean message here?

This part of the discussion shortened and streamlined.

(2) Line 643: is it also possible that the double inhibitor approach prevents the switch from homeostatic to reactive microglia instead of accelerating the opposite?

Our data do not support such mechanism, as we do see the activation of microglia 3 days post-injury in the inhibitor-treated animals. The major difference between the control and inhibitor treated animals are observed at 5 dpi corresponding to the period of the return to the homeostatic state (see distribution map in Fig. 7b). This discussion is included in the revised manuscript.

(3) Line 651: A coordinated response of astrocytes? What does this mean, please clarify.

We have streamlined the discussion part and also explained better the astrocyte reaction after the inhibitor treatment.

Figures:

(1) Figure 3: panel e: it would have been nice to see how well the cell types may cluster based on color pattern; for example, cluster 3 astrocytes and BAMs ? Was this analysed?

We now performed hierarchical clustering and aligned clusters according to their enrichment. Indeed, we observed that different cell types tended to cluster together and displayed enrichment at the same position along the spatial gradient. Surprisingly, despite their enrichment at the same spatial location, cluster 3_Astrocytes and 28_BAMs are not immediate neighbors in the clustering analysis. The new panel e is included in the revised figure.

(2) Figure 5: panels e-g: the cxcl10 signal in panel g is not convincing; please provide a better illustration.

We provided more convincing example as well as the quantification depicting the proportion of each cell type with detectable Cxcl10, Oasl2 and Ifi2712a signal. As this generated an additional large dataset, we decided to summarize this data in E.D. Fig. 8 in the revised manuscript.

(3) Figure 6: panels g, h: these panels are hard to read or interpret; the legend is also not helping a lot: either rework or remove; as is there is no point in keeping them in the main figures.

We agree with reviewer that these are difficult to follow. Fig. 6g and h in the revised manuscript displays, in our view, a better and more comprehensive representation of our findings.

(4) Figure 8 panels d,e and ED Fig 13: So from around 25k Edu+ cells about 6000 are oligs and 2500 astrocytes? What other cells are Edu Positive? MG? I am puzzled about the high percentage of Edu in Astrocytes (25 %!), this is in adult cortex; Is this only in the vicinity of the injury? Exactly when in time after the lesion? How much time was there for Edu incorporation? This finding deserves better discussion.

Part of the proliferating cells are microglia. However, the proliferating populations are not restricted to astrocytes, microglia and OPCs. Since EdU is incorporated during S-phase, we analyzed all proliferating cells in S-phase (based on a defined gene set expressed in S-phase) in response to injury using our single cell datasets. This is reported in the revised figure E.D. Fig. 5d.

We injected EdU 1hr prior to sacrifice and evaluated the proportion of EdU+ astrocytes in an area of 300 micrometers around the injury core (150 μ m on each side of the injury). The proportion of EdU-incorporating astrocytes that we observed in the control condition corresponds to previously published data (Frik et al. 2018) in stab wound model. All experimental details are reported in the materials and methods section.

(5) ED fig 1 panel d: the white dotted line for the lesion is hard to see; panel f: see earlier comments on similar graphs. Please check legend as well since the reference to panels appears wrong: should e not be d?

The figure and figure legend are modified according to the reviewer request.

(6) ED fig3: the title ‘Remaining’ clusters can have two meanings: Do you mean ‘Cells remaining in the lesion site/core’? Or ‘all other cells’? I guess you mean the latter, please clarify, and apply to other figures and legends.

We refer to the clusters that are not enriched at the injury site. In the revised figure, we refer to them as “all other clusters”.

(7) ED fig 6: I would have liked to see a panel for Aldh111.

Aldh111 is added to the revised figure.

(8) ED fig 7: Please rephrase the legend so that the content and meaning of the figure becomes more clear.

The figure legend is revised.

(9) ED fig 9: The figure is not really convincing, I also did not get it why here data for galectin-1 are shown; what is the plus?; Why is GFP in yellow in panel c? and IBA1 in blue?; Panel f: the clusters have no nr or name? to what other umap is this linked?

We added the galectin-1 as an example of the shared signature at the protein level. We felt that it would be important to show that our shared signature is not restricted to the mRNA, but also extends to the protein. As the data volume in the revised manuscript increased due to additional experiments we performed, we would follow the opinion of the reviewer that this panel is not essential for the manuscript and will not include it in the revised manuscript.

The panel f is in the revised manuscript in Figure 5 panel d and refers to the clusters of the UMAP in Figure 4b. This is clearly stated in the figure legend of the revised manuscript.

M&M:

(1) Line 1119: ‘operations’ should be ‘surgery’; please also add n to reveal nr of animals used in the study; and explain why only male mice are used;

We appreciate the valuable feedback provided by our reviewer. We have exchanged ‘operation’ with ‘surgeries’ and added the number of animals used to obtain our spatial- and single cell datasets in the revised methods and materials section. Given the previous reported sexual dimorphism in glial cell responses to pathologies (e.g., Villapol et al. 2017), we decided to exclusively focus our study on males to ensure data consistency and enable a targeted analysis within our predefined research scope. We added this statement in the revised method and material section.

Reviewer #3 (Remarks to the Author):

In this study, Koupourtidou and Schwarz et al. leveraged cutting-edge spatial transcriptomics approaches to provide comprehensive and unbiased spatial, temporal, and cell-type-specific profiling of gene expression in the murine cerebral cortex following a stab injury. From these studies, they uncovered an underappreciated injury-induced innate immunity-shared glial signature involving the Tlr1/2 and Cxcr3 signaling pathways. They next leveraged specific inhibitors/antagonists of Tlr1/2 and Cxcr3 in order to validate the functional relevance of these combined pathways in regulating glial responses in their stab model of traumatic brain injury (TBI). Here they found that combined abrogation of Tlr1/2 and Cxcr3 dampened the effects of stab injury on microglial morphology and astrocyte proliferation in the cerebral cortex. The research presented in this manuscript is novel, well-controlled, and presented in a manner that is both clear and provocative. While this is unquestionably a valuable resource for the TBI field, a few specific shortcomings and issues were noted, which I believe need to be addressed in order to solidify their central findings and conclusions.

Specific comments:

(1) The TLR1/2 signaling axis has been most extensively studied in the context of bacterial infection models where it has been shown to be a major orchestrator of innate immunity in response to bacterial lipoproteins. It would be interesting for them to speculate briefly somewhere in the text as to what they believe is driving TLR1/2 signaling in their stab TBI model. Most notably, it would be particularly interesting to know if they suspect that an endogenously-derived damage-associated molecular pattern (DAMP) is driving TLR1/2 activation in their model or if they believe that the stab puncture is now exposing the brain to bacterial ligands present on the skin surrounding the wound.

We have investigated the expression of known Tlr1/2 ligands and DAMPs. Indeed, we do observe their up-regulation in response to injury, compatible with the hypothesis that endogenously derived signal activates Tlr1/2. These data are now included in the revised manuscript.

(2) While their microglial morphology studies presented in Figure 7 are a good starting point to evaluate the effects of combined Cxcr3 and Tlr1/2 inhibition on microglial activation, additional studies are needed to strengthen their conclusions here. To this end, they should evaluate additional markers of microglial activation at the protein level to bolster their data interpretation here as changes in microglial morphology do not always provide an accurate quantification of microglial activation levels. Likewise, it would also help to boost the impact and significance of their spatial transcriptomics findings if they validated some of their major astrocyte findings in Figure 8 (e.g. changes in GFAP and LCN2 levels).

The morphological analysis of microglia is a standard in the field to assess their reactivity without relying on a specific marker (see for example: Heindl et al. 2018, Benakis et al. 2022). This was the reason to go for this analysis. In the revised analysis, we also included the quantification of RNA level (scRNAseq based) and protein levels (immunohistochemistry based)

for P2RY12, IBA1 and CD68. We do not observe any difference in the expression of these markers either on mRNA or protein levels between control and inhibitor treatment. In our opinion, these data suggest that use of single marker is not sufficient for the analysis. Moreover, this dataset further strengthens our analysis as we come to the same conclusion using scRNAseq and immunohistochemistry. These data are now included in the Fig. 7h-k and in the E.D. Fig. 14f-m.

Regarding the GFAP and Lcn2, we did stain for GFAP and NGAL (Lcn2) proteins and these data are shown in Fig. 1f and ED Fig. 15. First, the expression pattern does match the Visium data and the Lcn2 is shown in the Fig1f. For the validation of astrocyte reactivity, we analyzed the number of astrocytes expressing GFAP and NGAL in the control condition and after inhibitor treatment and do not observe change in number of cells expressing either GFAP or NGAL, despite the decrease of the expression levels caused by inhibitor treatment. This analysis was already presented in the ED Fig. 15 and fully validates our scRNAseq analysis. However, we did not explicitly point to it in our manuscript. We agree with our reviewer that this is an important point and it is now presented more clear in the revised manuscript. Moreover, this comment of our reviewer inspired us to check if the shared glial signature derived from the single cell analysis and downregulated after inhibitor treatment is enriched in the Visium dataset. Indeed, this is the case, further strengthening our spatial analysis. The plot demonstrating this enrichment is now included in the revised manuscript (E.D. Fig. 10).

(3) To increase the rigor of their findings presented in Figure 7f and Extend Data (ED) Figure 14a, they should provide average values per mouse as opposed to all acquisition data. Most notably, there appears to be data nesting in these figure panels (i.e. multiple data points from a low number of mice) and this could significantly skew and confound data interpretation.

Similar to the response to the previous point, the standard in the field for this analysis is to plot single cells (see for example Benakis et al. 2022). The reason for such analysis is that microglial features (e.g., soma volume, number of branches etc.) do not show normal distribution with the peak at one particular value. This is the consequence of the heterogeneous reaction of microglia to the insult. Therefore, it would not be correct to take the mean for one animal as a reliable parameter. This is the reason why we did not average the data per animal. However, to avoid the data nesting, we now analyzed the cell distribution for selected features and statistically compared the distributions between the control and inhibitor-treated animals. The new analysis fully support our previous conclusions and it is included in the revised manuscript (Fig. 7 + E.D. Fig. 14). In addition, we report the number of animals and number of cells analyzed per animal in the revised result and method section.

(4) Data quantification and statistical significance is needed to accompany the representative images presented in Figures 5 e-g and ED Fig 8 b-g.

We quantified the proportion of astrocytes, NG2 positive cells and microglia expressing Cxcl10, Oasl2 and Ifi2712a in the proximity of the injury. These data are included in the revised manuscript in ED Fig. 8.

Reviewer #4 (Remarks to the Author):

In this report the authors performed spatial and single cell transcriptomic analysis in the cortex of mice that received stab wound injury. They describe shared signatures of astrocytes microglia and oligodendrocytes at day 3 and 5 post injury. They aim to report functional relevance of the shared signatures however, as they state they primarily confirmed previous findings from their zebrafish models and other findings on injury to the brain. The conclusions are overstated, and the results presented here do not support the conclusions; there are no functional outcome measures. The data reported do not bring significant novel information that advance the field.

We disagree with the reviewer that our manuscript does not bring any novel information. Yes, we did study innate immunity in zebrafish, but only in the context of regulation of OPC proliferation. We did not observe any shared signature in the zebrafish. So, this is a new finding compared to our work in zebrafish. We even in the discussion of the current manuscript point out and discuss this discrepancy. Moreover, up to our knowledge, the shared signature of glial cells in response to injury has never been reported.

Similarly, we disagree with the judgment that our conclusion on the shared signature is overstatement. We provide the following evidence for it:

1. Single cell RNAseq analysis
2. Validation by RNAscope (mRNA-based validation)
3. Immunostaining validating the spatial transcriptomics
4. Functional data with pharmacological inhibition.

The major limitations are listed below:

(1) The transcriptomic data is both inconsistent and many details are left out. For the initial finding in Fig 3, it is a total of 6322 cells analyzed, then they subdivide this into 30 clusters. The number of cells per cluster is not given so we can't assess if this is even enough cells to make all of their additional conclusions (if you estimate an even split which obviously it is not you would say ~200 cells/cluster which is low, especially bc likely many of the clusters are less than this). They do not note how many animals are used for this analysis, given the low cell yield it seems like 1 per group which again is not sufficient to draw conclusions.

For every time point in our scRNAseq datasets, we collected tissue punches from 3 different animals (2 punches per animal as we perform bilateral injury). This is stated in the metadata submitted to the NCBI. However, indeed we did not comment on that directly in the method section. This is now included in the method section of the revised manuscript.

We felt that providing the number of cells for all cluster would be overloading the manuscript with data that might be not relevant for most readers. Instead, we will share the entire scRNA analysis notebook via GitHub along with publishing the manuscript and the GitHub link will be incorporated in the manuscript. This will allow everyone interested in details to look up not only the number of clusters and cells per cluster, but also the entire analysis with all parameters. We also prepared the table for reviewer that is submitted along with revised manuscript to allow

her/him to easily assess for each UMAP linked to our datasets the number of cells in each cluster per condition at this stage.

To address the reviewer criticism that our conclusions are based on too few cells and therefore not valid, we conducted a down-sampling experiment to further reduce the number of cells (for more details see below). This experiment did not adverse any of our previous findings and conclusions and further validated our initial analysis approach.

(2) In Figure 4, now they are comparing intact, 3 dpi, and 5 dpi, but it looks like the 3 dpi from Fig 3 is used, which isn't all that unusual except, in Fig 3 there are 3646 cells for 3 dpi and now in Fig 4 there are 3637, it is not noted why this number changes. Again, they do not list how many mice are added for each condition. Additionally, they subcluster the microglia, astrocytes, and oglios, but again don't list the total number of these cells to determine if they are powered enough to subcluster. Given these limitations, I would be hesitant to believe any of their single cell analysis (including spatial).

The discrepancy of nine cells in the 3dpi dataset resulted from our modification in doublet filtering, where we applied a strict doublet score during the combined analysis of all datasets. To address this discrepancy and maintain consistency, we conducted the entire analysis again using a consistent doublet score filter. This new analysis maintained the cell numbers constant throughout all analysis presented in the manuscript but did not change any conclusion presented in our initially submitted manuscript.

Regarding the number of cells, as stated above, we provided the table with cluster size to our reviewer to assess the power. Again, we do not find the cluster size critical as we do not base any conclusion on the absence of evidence that would suffer from the low cell number. All our conclusions are based on the positive evidence.

Regarding the reliability of our analysis, we can only further stress that we validate our major conclusions using RNAscope, immunohistochemical analysis and functional readouts. These validations, in our opinion, rather make our analysis very rigorous and reliable (as also stated by other reviewers).

(3) They analyzed 16567 cells in intact brains, 3637 at 3dpi and 13658 at 5dpi, why there is 5 fold less cells at 3dpi? Same for the analysis with the treatments (Cx3Cr1 and Tlr2) they analyzed 16649 from intact animals and 3643 from 3dpi 4613 and 13766 and 15142 from 5dpi. This is a major caveat to the interpretation of the results.

In order to be able to perform proper comparison between different experiments, we always included the intact sample that was used for the appropriate batch correction. This is the reason for large number of cells in the intact condition. To exclude the possibility that the different number of cells influences our conclusions, we down-sampled all datasets to the size of the smallest data set using `sc.pp.subsample()` as a down sampling method. This did not alter either clusters that we observed or the differentially expressed genes and enriched GO terms. This analysis is included to the submission of the revised manuscript as figure for our reviewers.

(4) In Figure 5 they compare their dataset with an LPS injection model, but not published TBI datasets? It is unclear why they would not compare with published TBI datasets. In the discussion

they mentioned that the compared “the signature that are also present in other brain pathologies such as LPS” (line 585) . LPS is not a brain pathology.

We agree with the statement that LPS pathology is not precise enough. This will be rephrased to LPS-induced neuroinflammation in the revised manuscript. Furthermore, we conducted a comparative analysis with datasets from Arneson et al. for FPI (fluid percussion injury) and Zheng et al. for MCAO (middle cerebral artery occlusion) mouse models. This observation adds an intriguing dimension to our study, emphasizing the dynamic nature of the shared signature in glial reactivity. We have incorporated this noteworthy comparison into the revised manuscript, and we express our gratitude to the reviewer for highlighting this aspect.

(5) For Fig 7 - it is not clear how many mice are analyzed per group. The figure legend notes 450 microglia per condition, but not the number of mice per condition. Additionally, Fig 8 has the number of mice graphed where as Fig 7 has the number of cells, why is there this inconsistency?

For the microglia morphology analysis (Fig. 7), we used 6 animals for control and 5 for inhibitor treatment. This is now stated in the method section of the revised manuscript. As outlined in response to reviewer 1, morphological microglial features are generally plotted as single cells to display the cellular heterogeneity of microglial reaction. Expressing the average per animal would result in a loss of information. In contrast, the astrocyte proliferation in response to injury (Fig. 8) shows the binary response (cells either proliferate or not). Therefore, it is possible to express the proportion of cells that proliferate in response to injury per single animal without losing any information.

(6) For Fig. 9 when they add in the antagonist, they need to clarify if they are adding both the Cxcr3 and Tlr1/2 antagonist together. Why they did not test adding them independently? Secondly, they confirm the chemical compounds in an OPC knock-out cell line, why not microglia or astrocyte? Again it looks like they are using the analysis from Fig 3 and 4, but now 3 dpi has 3643 cells (so different from Fig 3 and Fig 4). Additionally, the 3 dpi + INH only has 4613 cells, whereas 5 dpi has 15142 - why is there this large difference? Again the number of mice per group is not noted.

The observed difference between Fig. 3+4 and Fig. 9 comprises 3 cells. As pointed out above, we used in the revised manuscript the same filtering to avoid these small differences. Similarly, as pointed out above, the down-sampling analysis shows that with reduced number of cells we still attain the same results.

For the inhibitors, we inhibited both pathways simultaneously as in our previous zebrafish work only the dual- and not the single pathway inhibition resulted in reduced glial reactivity. In the revised manuscript, we provided more details on the inhibitor treatment (materials and method section) to fully address our reviewers point. The validation of inhibitors is done in our published work, and it is not part of this manuscript. We specified this more explicitly.

(7) The injury used is a stab wound injury that does not recapitulate fully the chronic TBI consequences as they themselves state for instance in the resolution of some of the signatures by 5dpi. The manuscript is misleading as it referred to TBI while this is a very specific kind of injury and should be specified:

In the title (line 1), Results (Line 92, 242) As a result the intro should be tailored to this and not to all TBI.

The stab wound injury is a particular type of traumatic brain injury caused by the penetrating object. We, however, agree with our reviewer that this is a particular type of injury and re-phrased TBI to stab wound injury (SWI) in the revised manuscript.

(8) They only use male mice (not clear how many) it is not justify why female mice were not included.

We have added the animal number used to obtain our spatial- and single cell datasets in the revised material and method section. Considering that glial cells react differently to pathologies in males and females (see for example Villapol et al. 2017), we have decided to focus our study on male mice. We believe that a comparison of male and female mice is beyond the scope of this manuscript. In the revised manuscript's method section, we included our restriction to male mice.

(9) Line 58: Activated microglia should be changed with Reactive Microglia as it is well accepted that we can't describe microglia as "activated".

In the revised manuscript, we use term reactive microglia.

(10) "Sacrificed" should always be replaced with Euthanized.

We agree with reviewer and replaced "sacrificed" with "euthanized".

REVIEWERS' COMMENTS

Reviewer #1 (Remarks to the Author):

In this revision the authors have fully address the concerns that I had. I would recommend it to be accepted by Nature Communications.

Reviewer #2 (Remarks to the Author):

Upon reading the revision of the manuscript I am satisfied with all the changes made. Overall the quality of the papers benefits from the adaptations. This work will be of great value to the research community.

There is only one small suggestion left from my side:

Please make clear in the M&M section or the figure legend or the text that NGAI protein stainings relate to lcn2 gene epxression. This is not clearly stated in the current version of the manuscript.

Reviewer #3 (Remarks to the Author):

The authors have done a fine job of addressing my comments and critiques.

Reviewer #4 (Remarks to the Author):

the authors addressed all the comments

Point-by-point response to the reviewers' comments:

Reviewer 1:

No comments after revision.

Reviewer 2:

No comments after revision.

Reviewer 3:

There is only one small suggestion left from my side: Please make clear in the M&M section or the figure legend or the text that NGA1 protein stainings relate to lcn2 gene expression. This is not clearly stated in the current version of the manuscript.

We included the clarification for the NGA1 staining in the Method section.

Reviewer 4:

No comments after revision.